# Biochemical and neurophysiological effects of deficiency of the mitochondrial import protein TIMM50

**Eyal Paz**[1,2†], **Sahil Jain**[1†], **Irit Gottfried**[1], **Orna Staretz-Chacham**[3], **Muhammad Mahajnah**[4,5], **Pritha Bagchi**[6,7,8], **Nicholas T Seyfried**[6,7], **Uri Ashery**[1,2*], **Abdussalam Azem**[1,2*]

[1]School of Neurobiology, Biochemistry and Biophysics, George S. Wise Faculty of Life Sciences, Tel Aviv University, Tel Aviv, Israel; [2]Sagol School of Neuroscience, Tel Aviv University, Tel Aviv, Israel; [3]Metabolic Disease Unit, Soroka Medical Center, Faculty of Health Sciences, Ben-Gurion University of the Negev, Beersheba, Israel; [4]The Ruth and Bruce Rappaport Faculty of Medicine, Technion-Israel Institute of Technology, Haifa, Israel; [5]Child Neurology and Development Center, Hillel Yaffe Medical Center, Hadera, Israel; [6]Center for Neurodegenerative Diseases, Emory University, Atlanta, United States; [7]Department of Biochemistry, Emory University, Atlanta, United States; [8]Emory Integrated Proteomics Core, Emory University, Atlanta, United States

**\*For correspondence:**
uriashery@gmail.com (UA);
azema@tauex.tau.ac.il (AA)

†These authors contributed equally to this work

**Competing interest:** The authors declare that no competing interests exist.

## eLife Assessment

This **important** study presents interesting results aimed at explaining the effects of a human mutation on the mitochondrial import protein TIMM50 on mitochondrial function and neuronal excitability. While the evidence supporting the conclusions is **convincing**, the mechanisms driving changes in the levels of certain proteins within and outside the mitochondria (such as certain ion channels) remain unexplained. This paper will be of interest to scientists in the mitochondria field.

**Abstract** TIMM50, an essential TIM23 complex subunit, is suggested to facilitate the import of ~60% of the mitochondrial proteome. In this study, we characterized a *TIMM50* disease-causing mutation in human fibroblasts and noted significant decreases in TIM23 core protein levels (TIMM50, TIMM17A/B, and TIMM23). Strikingly, TIMM50 deficiency had no impact on the steady-state levels of most of its putative substrates, suggesting that even low levels of a functional TIM23 complex are sufficient to maintain the majority of TIM23 complex-dependent mitochondrial proteome. As TIMM50 mutations have been linked to severe neurological phenotypes, we aimed to characterize TIMM50 defects in manipulated mammalian neurons. TIMM50 knockdown in mouse neurons had a minor effect on the steady state level of most of the mitochondrial proteome, supporting the results observed in patient fibroblasts. Amongst the few affected TIM23 substrates, a decrease in the steady state level of components of the intricate oxidative phosphorylation and mitochondrial ribosome complexes was evident. This led to declined respiration rates in fibroblasts and neurons, reduced cellular ATP levels, and defective mitochondrial trafficking in neuronal processes, possibly contributing to the developmental defects observed in patients with TIMM50 disease. Finally, increased electrical activity was observed in TIMM50 deficient mice neuronal cells, which correlated with reduced levels of KCNJ10 and KCNA2 plasma membrane potassium channels, likely underlying the patients' epileptic phenotype.

## Introduction

The mitochondrion is a vital organelle found in nearly all eukaryotic cells, where it is involved in numerous important cellular functions and metabolic pathways, including supplying cellular energy, assembling iron-sulphur clusters, and regulating the cell cycle, cell growth, and differentiation, programmed cell death, and synaptic transmission (*Reichert and Neupert, 2004*; *Stehling et al., 2014*; *Pfanner et al., 2019*). In humans, these functions are executed by ~1500 different mitochondrial proteins, of which only 13 are encoded by the mitochondrial genome (*Filograna et al., 2021*). The remaining mitochondrial proteins are nuclear-encoded and thus imported into the mitochondria. The translocated proteins are sorted to their specific mitochondrial compartment via several intricate protein translocation pathways, including the presequence pathway, which is used for the import of nearly ~60% of the mitochondrial proteins (*Schmidt et al., 2010*; *Pfanner, 2016*).

The TIM23 complex mediates the import of some intermembrane space (IMS) proteins, many mitochondrial inner membrane (MIM) proteins, and all mitochondrial matrix proteins (*Mokranjac and Neupert, 2010*). The TIM23 complex in yeast comprises three essential subunits, Tim23, Tim17, and Tim50 (TIMM23, TIMM17A/B, and TIMM50 in mammals). Association of Tim21 (TIMM21 in mammals) and Mgr2 (ROMO1 in mammals) promotes the lateral translocation of proteins into the MIM, while the association of the presequence translocase-associated motor (PAM) complex with the TIM23 core promotes the import of matrix proteins (*Mokranjac et al., 2005*; *Ieva et al., 2014*; *Richter et al., 2019*). Recent structural analysis showed that Tim17 forms the protein translocation path, whereas the associated Tim23 protein likely plays a structural role, serving as a platform that mediates the association of other complex subunits (*Sim et al., 2023*; *Fielden et al., 2023*).

Tim50, first discovered in yeast some two decades ago (*Geissler et al., 2002*; *Mokranjac et al., 2003*), is thought to be the first TIM23 complex component to interact with presequences of precursor proteins as they emerge from the Tom40 channel, thus playing a pivotal role in presequence-containing protein sorting (*Yamamoto et al., 2002*). It was further suggested that normal Tim50 functionality is required for maintaining the mitochondrial membrane potential (*Meinecke et al., 2006*). Additionally, TIMM50, the mammalian homologue of Tim50, was shown to be involved in steroidogenesis and plays a preventive role in pathological cardiac hypertrophy and several types of cancer (*Bose et al., 2019*; *Tang et al., 2017*; *Sankala et al., 2011*; *Gao et al., 2016*; *Zhang et al., 2019*).

Recently, TIMM50 has generated immense interest in human health research, as mutations in the encoding gene have been linked in geographically and ethnically varied populations to the development of a severe disease characterized by mitochondrial epileptic encephalopathy, developmental delay, optic atrophy, cardiomyopathy, and 3-methylglutaconic aciduria. To date, seven different mutations have been identified in children from ten unrelated families (*Serajee, 2015*; *Shahrour et al., 2017*; *Tort et al., 2019*; *Reyes et al., 2018*; *Mir et al., 2020*; *Moudi et al., 2022*).

Although TIMM50 mutants are mostly associated with neurological disorders, functional characterization of TIMM50 has yet to be reported in neuronal cells. Additionally, despite being involved in the import of nearly 60% of the mitochondrial proteins, the impact of TIMM50 deficiency on the entire mitochondrial proteome has yet to be characterized. In this study, using a proteomics approach, we show the impact of TIMM50 deficiency on the mitochondrial and cellular proteome and characterize, for the first time, the neurological role of TIMM50 by knocking it down in mouse neurons.

## Results

### *TIMM50* disease-causing mutation reduces the levels of TIM23 complex components in patient fibroblasts

A previous study reported the clinical features of a rare genetic disease in several patients from two unrelated Arab families residing in Israel and Palestine (*Shahrour et al., 2017*). Patients suffering from the disease exhibited various neurological and metabolic disorders, including epilepsy, developmental delay, optic atrophy, cardiomyopathy, and 3-methylglutaconic aciduria. The disease was linked to a homozygous missense mutation resulting in T149M replacement in the *TIMM50* gene (*Figure 1A*). To unravel the molecular basis of the disease, we examined the effect of the mutation on the proteome of primary fibroblasts collected from two affected family members (P1 and P2), as well as from a healthy relative (HC).

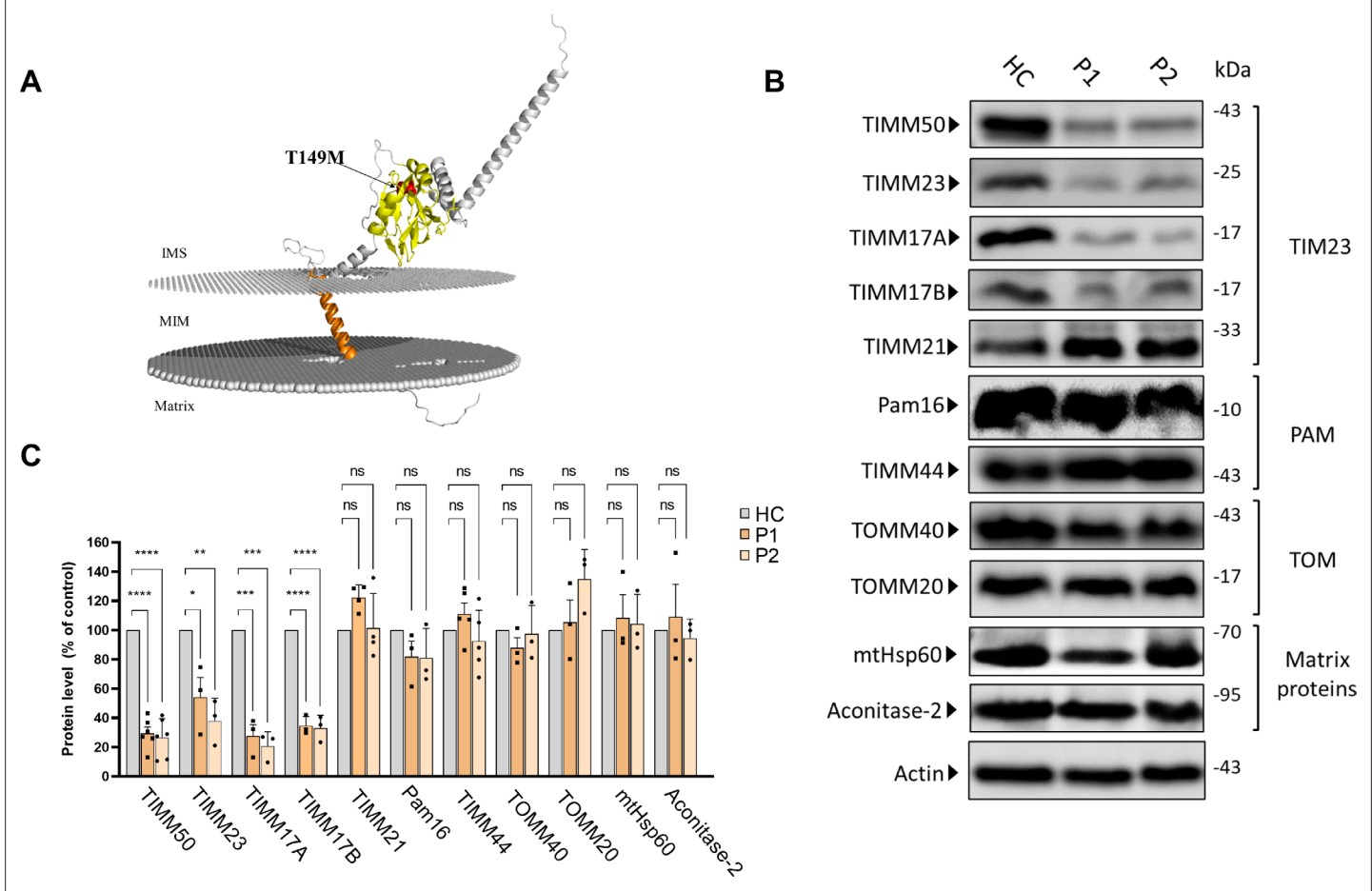

**Figure 1.** Differential effects of *TIMM50* mutation on the expression of TIM23, presequence translocase-associated motor (PAM) and TOM subunits and matrix-destined proteins. (**A**) Predicted AlphaFold (*Tunyasuvunakool et al., 2021*) human TIMM50 structure displaying the position of the T149M mutation. PPM webserver (*Lomize et al., 2022*) was used to embed the predicted structure into the mitochondrial inner membrane (MIM). Orange – transmembrane domain, Yellow – FCP1-like domain, Red – Threonine 149. (**B**) Healthy control- and patients-derived primary fibroblasts were lysed and analyzed by immunoblot with the indicated antibodies. Actin was used as the loading control. Full results and original blots are found in *Figure 1— source data 1*; *Figure 1—source data 2*. (**C**) Band density analysis of the blots presented in A and in *Figure 1—source data 1*; *Figure 1—source data 2*, showing significant decrease in levels of TIMM50, TIMM23, and TIMM17A/B, but not in TIMM21, PAM subunits, TOM subunits, and matrix TIM23 complex substrates. Band signals were normalized to the loading control and compared to the level measured in the healthy control sample, taken as 100%. Data are shown as means ± SEM, n=6 biological repeats for TIMM50 antibody, n=3–5 biological repeats for all other antibodies, *p-value <0.05, **p-value <0.01, ***p-value <0.001, ****p-value <0.0001, Ordinary one-way ANOVA.

The online version of this article includes the following source data for figure 1:

**Source data 1.** PDF file containing original immunoblots for *Figure 1*, indicating the relevant bands and samples running order.

**Source data 2.** Original files for immunoblot analysis are displayed in *Figure 1*.

We first examined the effect of the mutation on TIMM50 levels by immunoblotting and found that the mutation leads to a significant decrease in TIMM50 levels in patient fibroblasts (*Figure 1B and C*). We next examined the effect of the *TIMM50* mutation on other components of the TIM23 complex. Notably, our analysis revealed a significant reduction in the levels of the core TIM23 complex subunits, namely, TIMM23, TIMM17A, and TIMM17B (*Figure 1B and C*). These results are in agreement with previous reports showing that two other *TIMM50* mutations (resulting in R217Q+G372 S and S112*+G190 A replacements, both compound heterozygous) also led to major decreases in TIMM50, TIMM23, and TIMM17A/B levels (*Tort et al., 2019*; *Reyes et al., 2018*).

In contrast to the decreased levels of TIM23 complex core components seen in the patient fibroblasts, the levels of subunits belonging to the PAM complex were not affected, despite their expected import dependency on the TIM23 complex. TIMM21 also showed minimal to no change in amount.

Additionally, as subunits of the TOM complex that serves as the general import pore in the outer membrane were shown to interact with TIM23 complex subunits, including TIMM50 (*Shiota et al., 2011*), we also considered the possible effect of reduced TIMM50 levels on the expression of TOM complex subunits. No changes in the levels of TOM complex subunits addressed in patient fibroblasts were noted (*Figure 1B and C*).

Finally, as TIM23 is known to be the sole gateway into the mitochondrial matrix (*Mokranjac and Neupert, 2010*), we examined the effect of *TIMM50* mutation on the steady state levels of the matrix proteins aconitase 2 and mtHsp60. Surprisingly, the steady state levels of both proteins were unchanged in patient fibroblasts (*Figure 1A and B*), despite the significant loss of TIMM50, TIMM23, and TIMM17A/B.

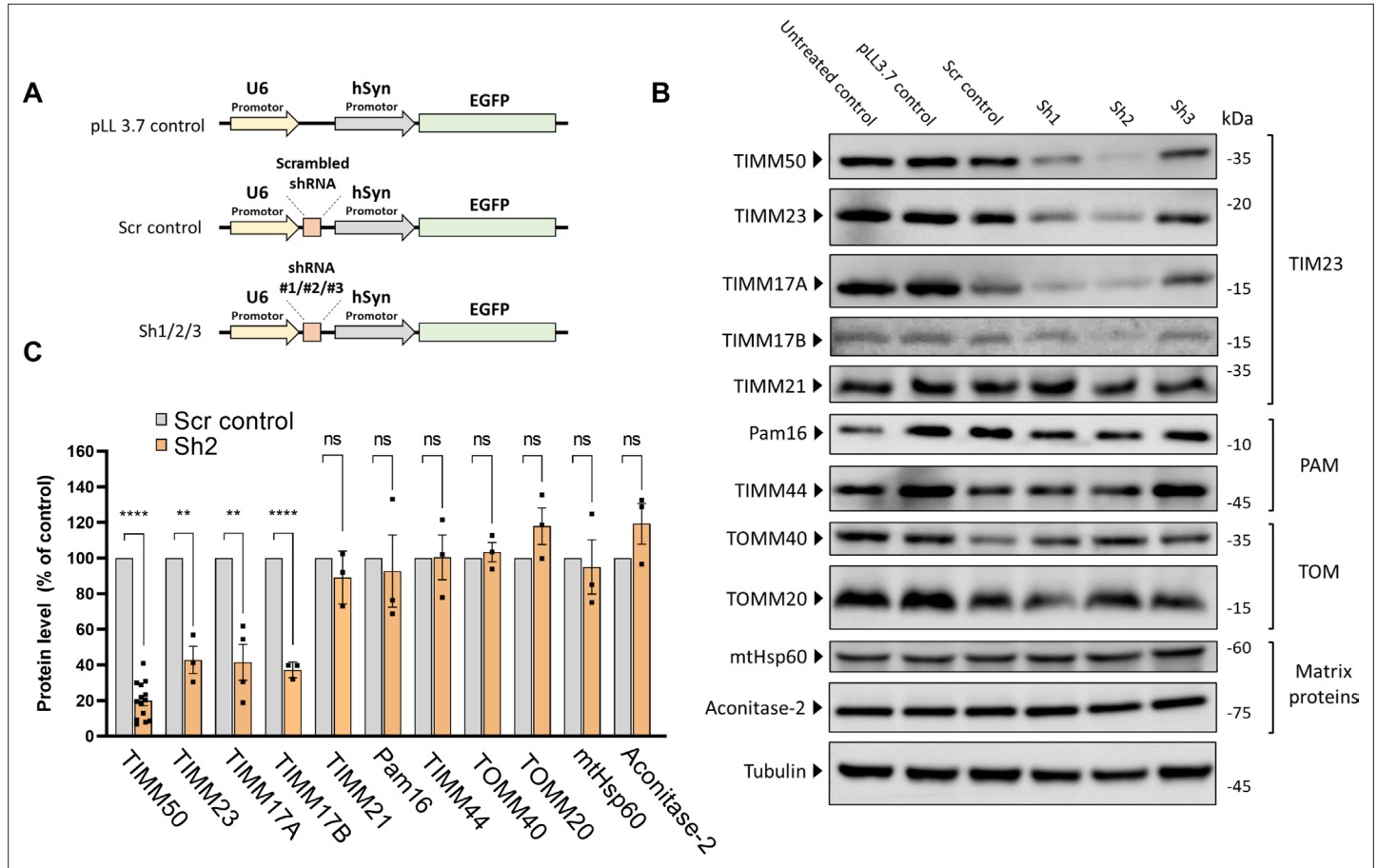

**Figure 2.** TIMM50 knockdown in mouse primary cortical neurons serves as a model system to study TIMM50 deficiency in mammalian neurons. (**A**) Schematic depictions of the different constructs used in this study. 'pLL3.7 control' is a control for EGFP expression. 'scrambled (Scr) control' is a control for the shRNA system activation with a scrambled Sh3 sequence and Sh1-Sh3 are three plasmids expressing different TIMM50 targeting shRNA sequences. (**B**) Neuronal cultures were transduced to express the indicated constructs, lysed, and analyzed by immunoblot with the indicated antibodies. Tubulin was used as the loading control. 'Untreated' controls are cells that were not transduced. Full results and original blots are found in *Figure 2—source data 1*; *Figure 2—source data 2*. (**C**) Band density analysis of the blots presented in B (and in *Figure 2—source data 1*; *Figure 2—source data 2*) shows a significant decrease in levels of TIMM50, TIMM23, and TIMM17A/B, but not in TIMM21, presequence translocase-associated motor (PAM) subunits, TOM subunits, and matrix TIM23 complex substrates. Band signals were normalized to the loading control and compared to the Scr control, taken as 100%. Data are shown as means ± SEM, n=14 biological repeats for TIMM50 antibody, n=3–4 biological repeats for all other antibodies, **p-value <0.01, ****p-value <0.0001, unpaired Student's t-test.

The online version of this article includes the following source data for figure 2:

**Source data 1.** PDF file containing original immunoblots for *Figure 2*, indicating the relevant bands and samples running order.

**Source data 2.** Original files for immunoblot analysis are displayed in *Figure 2*.

## Generating a neuronal model system to study the neurophysiological effects of TIMM50 deficiency

As *TIMM50* mutations lead to severe neurodevelopmental symptoms and to a significant reduction in steady-state TIMM50 levels (*Figure 1* and *Tort et al., 2019*; *Reyes et al., 2018*) we wanted to study the effects of TIMM50 deficiency in neuronal cells by knocking down TIMM50 in mouse primary cortical neurons. For this purpose, we designed three shRNA sequences (termed Sh1, Sh2, and Sh3; vector schemes are presented in *Figure 2A*) and cloned them into a lentiviral vector that also allows for EGFP expression under the control of the hSyn promotor (*Ben-Simon et al., 2015*). These vectors allow TIMM50 knockdown (KD) while specifically labeling neurons with EGFP, allowing for efficient visualization in single-cell experiments. We compared TIMM50 expression relative to three controls, namely, an untreated control (i.e. cultures that were not transduced), a pLL3.7 control (i.e. cultures that were transduced but did not transcribe the shRNA sequence, yet expressed EGFP), and a shRNA system activation control (i.e. cultures that were transduced with a scrambled shRNA sequence). All the three targeting shRNA sequences had an impact on TIMM50 levels, in comparison to the controls. Sh2 had the most significant and consistent effect, reducing TIMM50 levels by ~80% (*Figure 2B and C*). Therefore, Sh2 was chosen as the KD vector for all subsequent experiments on neurons.

We next examined the effects of TIMM50 KD in neurons, addressing the same components as were tested in fibroblasts. In complete agreement with the fibroblast results, the levels of TIMM23 and TIMM17A/B were significantly decreased, while the amounts of TIMM21, PAM subunits, TOM subunits, and the matrix substrates tested were not affected (*Figure 2B and C*). Hence, similarly to what was detected in fibroblasts, steady-state levels of several representative mitochondrial proteins were not affected in TIMM50 KD neurons.

## TIMM50 deficiency does not affect the steady-state levels of a majority of its substrates

It was expected that the significant decrease in the levels of TIM23 core components seen upon TIMM50 deficiency would decrease the levels of substrates processed by this translocation complex. The observation that steady-state levels of two mitochondrial matrix substrates, mtHsp60 and aconitase 2, were not affected by the significant loss of TIM23 core components in both research systems (*Figures 1A, B ,, 2B and C*), motivated us to examine the impact of TIMM50 KD on the total mitochondrial and cellular proteomes. For this purpose, we performed untargeted mass spectrometry analysis of fibroblasts from both patients, as compared to the healthy control, and of mice primary neurons transduced with either Sh2 or the Scr control. For fibroblasts, principal component analysis (PCA) was performed to verify that the replicates of both patients were consistently different from the HC replicates (*Figure 3—figure supplement 1*).

In the case of fibroblasts, 127 MIM and 190 matrix proteins were detected using mass spectrometry (*Source data 1 and 2*). Surprisingly, we noticed that the levels of 83 (~65%) MIM proteins and 135 (~71%) matrix proteins were not affected in either patient, as compared to the HC (*Figure 3A and B* and *Source data 1 and 2*). Among the MIM proteins that were not affected by the *TIMM50* mutation, we identified multiple proteins involved in calcium homeostasis (such as MICU2, SLC25A3, and LETM1), heme synthesis (such as PPOX and CPOX), and cardiolipin synthesis (HADHA). Among matrix proteins that were not affected by *TIMM50* mutation, we identified multiple proteins involved in Fe-S cluster biosynthesis (such as NFS1, GLRX5, and ISCU), detoxification (such as PRDX5, SOD2, ABHD10, and GSTK1), fatty acid oxidation (such as DECR1, ECHS1 and ETFA), and amino acid metabolism (PYCR1, ALDH18A1, and HIBCH). Also, the majority of TCA cycle proteins, such as ACO2, DLST, IDH3B, and OGDH, found in the matrix, were not affected in patient fibroblasts. Interestingly, every MIM protein that changed significantly in both patients, appeared to be downregulated, while the significantly changed matrix proteins comprised of both downregulated and upregulated proteins (*Figure 3C*). Unexpectedly, a few matrix proteins (ALDH2, GRSF-1, AK4, LACTB2, and OAT) showed increased steady-state levels in both patients (*Figure 3C*). Of these proteins, ALDH2 and GRSF-1 showed ~15 and ~sixfold increases, respectively, as was also confirmed by immunoblot (*Figure 3—figure supplement 2*).

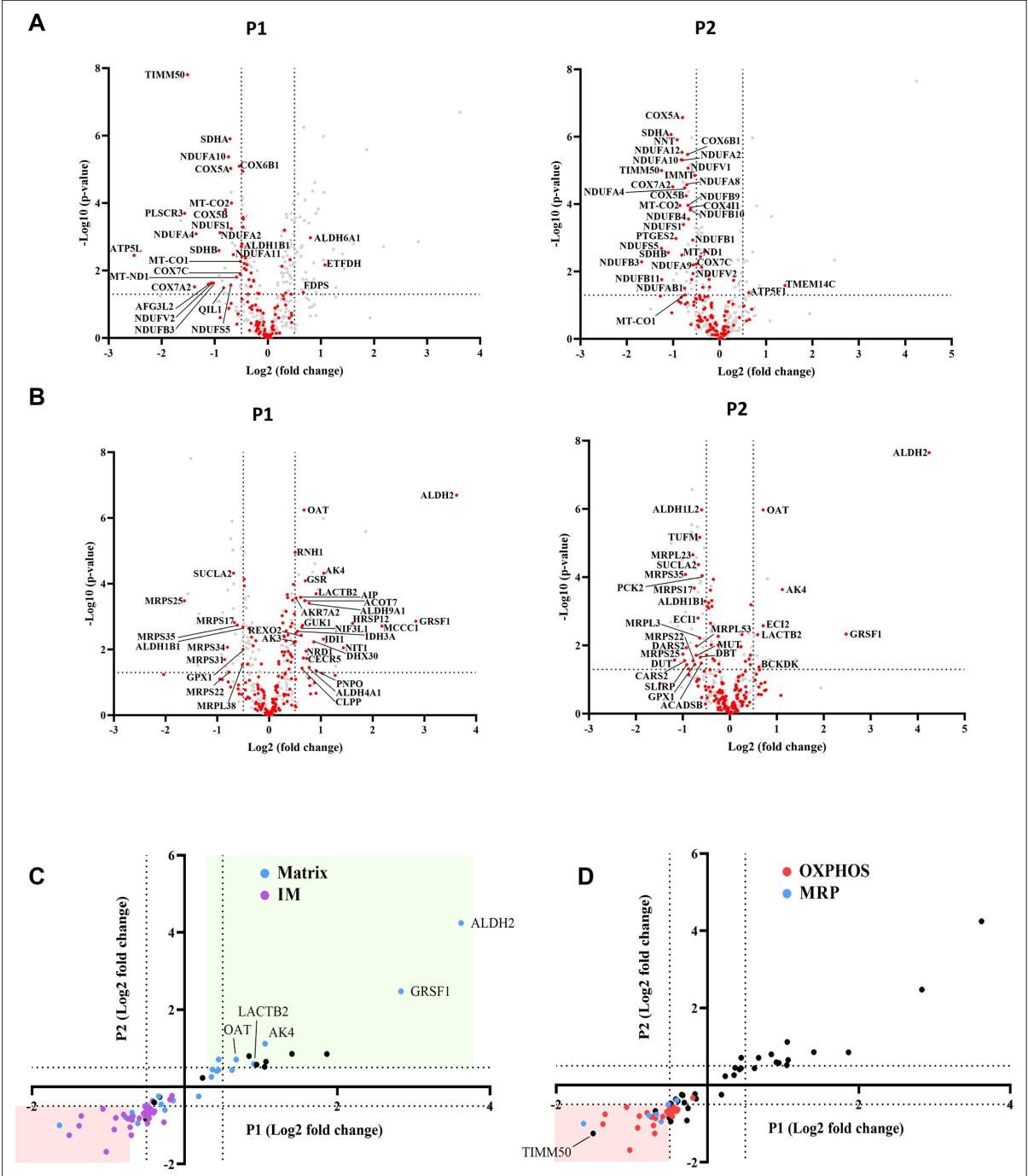

**Figure 3.** TIMM50 deficiency leads to a specific reduction in the oxidative phosphorylation (OXPHOS) and mitochondrial ribosomal proteins (MRP) machineries but does not affect the majority of mitochondrial inner membrane (MIM) and mitochondrial matrix proteome. Untargeted mass spectrometry analysis of fibroblasts from both patients (P1, P2), as compared to the healthy control. In the volcano plots displayed in (**A, B**) and in all subsequent figures, the y-axis cut-off of >1.301 corresponds to –log (0.05) or p-value = 0.05, while the x-axis cut-off of <−0.5 and >0.5 corresponds to a ±1.414 fold change. Each dot in the graph represents a protein. Proteins depicted on the right side of the x-axis cut-off and above the y-axis cut-off were considered to be increased in amount, while proteins depicted on the left side of the x-axis cut-off and above the y-axis cut off were considered to be decreased in amount. All relevant proteins that were increased/decreased in amount are identified by their gene name, next to the dot. Statistical analysis was performed using Student's t-test and a p-value <0.05 was considered statistically significant. n=9 per group (three biological repeats in triplicate), P1 and P2 results were compared to healthy control (HC) results. Full list of differentially expressed proteins in fibroblasts is found in *Source data 1 and 2*. (**A**) The steady-state levels of a majority of MIM proteins detected in patient fibroblasts were not affected. MIM proteins are colored red, while other detected mitochondrial proteins are colored gray. (**B**) The steady-state levels of a majority of matrix proteins detected in patient fibroblasts

*Figure 3 continued on next page*

*Figure 3 continued*

were not affected. Matrix proteins are colored red, while other detected mitochondrial proteins are colored gray. (**C**) Correlation plot comparing the log-fold-change of P1 to that of P2 showing that out of the proteins that were significantly changed for both patients, MIM proteins are consistently downregulated, while matrix proteins are both upregulated and downregulated. (**D**) Correlation plot comparing the log-fold-change of P1 to that of P2 showing that OXPHOS and MRP proteins comprise nearly all the proteins that were downregulated in both patients.

The online version of this article includes the following source data and figure supplement(s) for figure 3:

**Figure supplement 1.** Principal component analysis (PCA) analysis for fibroblasts proteomics replicates.

**Figure supplement 2.** Immunoblot confirmation of mass spectrometry-based proteomics findings for ALDH2 and GRSF1.

**Figure supplement 2—source data 1.** PDF file containing original immunoblots for *Figure 3—figure supplement 2*, indicating the relevant bands and samples running order.

**Figure supplement 2—source data 2.** Original files for immunoblot analysis are displayed in *Figure 3—figure supplement 2*.

**Figure supplement 3.** Go-term analysis of all significantly decreased fibroblast proteins and OXA1 immunoblot.

**Figure supplement 3—source data 1.** PDF file containing original immunoblots for *Figure 3—figure supplement 3*, indicating the relevant bands and samples running order.

**Figure supplement 3—source data 2.** Original files for immunoblot analysis are displayed in *Figure 3—figure supplement 3*.

## Oxidative phosphorylation subunits and mitochondrial ribosomal proteins comprise the majority of down-regulated proteins in TIMM50-deficient fibroblasts

A total of 69 oxidative phosphorylation (OXPHOS) and 27 mitochondrial ribosomal proteins (MRP) were detected in fibroblasts by mass spectrometry. Remarkably, of the 18 MIM proteins found in decreased amounts in both patients, 17 belong to the OXPHOS family, while of the seven matrix proteins found in decreased amounts in both patients, four proteins belong to the MRP family (*Figure 3D*). The OXPHOS complexes most affected were CI, CII, and CIV. In the case of CI, one membrane core subunit (MT-ND1), two hydrophilic core subunits (NDUFS1 and NDUFV2), and four super-numerary subunits (NDUFA2, NDUFA10, NDUFB3, and NDUFS5) exhibited a significant negative fold-change in both patients (*Source data 1* and *Source data 2*). The levels of two CII subunits, namely, SDHA and SDHB, and eight CIV subunits, including the two catalytic subunits MT-CO1 and MT-CO2, were also significantly decreased in both patients. GO term analysis performed on all the genes that were downregulated in both patients, confirmed enrichments of mitochondrial inner membrane proteins, several OXPHOS-related processes, and mitochondrial ribosome subunits (*Figure 3—figure supplement 3A*).

Interestingly, as mentioned above, several mitochondrially encoded OXPHOS subunits were found to be downregulated. These subunits are translated by the mitochondrial ribosomes and assembled into the MIM by the oxidase assembly (OXA) insertase (*Itoh et al., 2021*). Therefore, these subunits are not directly related to TIM23 import. To elucidate the reason behind the downregulation of these proteins, we tested the levels of the OXA insertase by immunoblotting and found no significant difference in its levels between the patient fibroblasts and the HC (*Figure 3—figure supplement 3B* and C). Therefore, we conclude that the decreased levels of MRP subunits likely lead to lower translation rates of mitochondrially encoded OXPHOS subunits and consequently, to their lower abundance in the MIM.

## The impact of TIMM50 deficiency on the mitochondrial proteome in neurons

In TIMM50 KD neurons, 170 MIM and 215 matrix proteins were detected by mass spectrometry (*Source data 3*). Similar to the observations for fibroblasts, the majority of the MIM and matrix proteins were not affected by TIMM50 KD (*Figure 4A and B*). OXPHOS and MRP subunits showed a similar trend to what was observed in patient fibroblasts and appeared to be mostly downregulated, although the affected OXPHOS subunits appeared to decrease to a lesser extent (*Figure 4C and D*). The difference in the extent of fold-change in neurons, as compared to patient fibroblasts, could be due to the short duration of KD in the neuronal experiment, as compared to that with the patient fibroblasts, that constantly carry the deficiency.

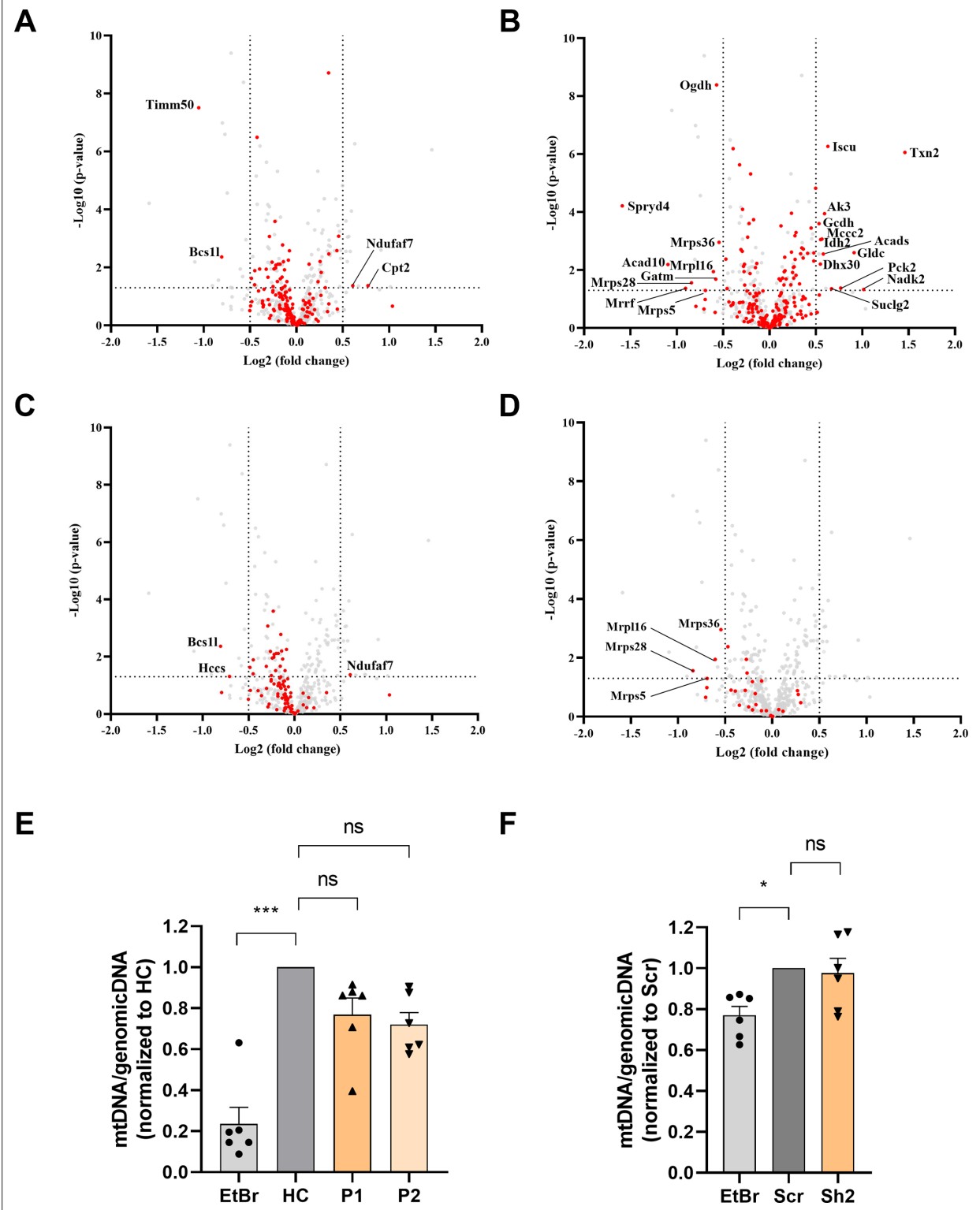

**Figure 4.** The impact of TIMM50 deficiency on the mitochondrial proteome in neurons. Untargeted mass spectrometry analysis of mice primary neurons transduced with either Sh2 or the scrambled (Scr) control. n=9 per group (three biological repeats in triplicates). Full list of differentially expressed proteins in neurons is found in ***Source data 3***. (**A**) The steady-state levels of a majority of mitochondrial inner membrane (MIM) proteins detected in TIMM50 knockdown (KD) neuronal cells were not affected. MIM proteins are colored red, while other detected mitochondrial proteins are colored gray. (**B**) The steady-state levels of a majority of matrix proteins detected in TIMM50 KD neuronal cells were not affected. Matrix proteins are colored red, while other detected mitochondrial proteins are colored gray. (**C**) Most OXPHOS subunits are observed to be in decreased amounts in TIMM50

*Figure 4 continued on next page*

*Figure 4 continued*

KD neuronal cells, albeit at a lower extend from what was observed in patient fibroblasts. OXPHOS proteins are colored red, while other detected mitochondrial proteins are colored gray. (**D**) Decreased steady-state levels of multiple mitochondrial ribosomal protein (MRP) subunits were observed in TIMM50 KD neuronal cells. MRP proteins are colored red, while other detected mitochondrial proteins are colored gray. (**E, F**) Mitochondrial DNA content in fibroblasts (**E**) and neurons (**F**) was not affected in TIMM50 deficient cells compared to control cells. Mitochondrial DNA content was estimated by measuring the ratio between mitochondrial and nuclear DNA. Dloop1 expression was measured by qPCR relative to hypoxanthine guanine phosphoribosyl transferase (HPRT); TERT served as a control of a nuclear-encoded gene. Ethidium bromide (EtBr; 100 ng/ml) was used as the positive control. Data are shown as means ± SEM, n=6 (samples from three biological replicates, each performed twice), *p-value <0.05, ***p-value <0.001, Kruskal-Wallis test.

Additionally, to determine whether the changes observed in the levels of OXPHOS and MRP proteins were an indirect effect resulting from general mitochondrial DNA loss, we performed a qPCR assessment of the mitochondrial DNA content. Our results showed no effect on mitochondrial DNA levels in either model system (*Figure 4E and F*), thereby confirming that the observed effects on the OXPHOS and MRP protein levels were the direct and specific result of TIMM50 deficiency, and not indirectly due to mitochondrial DNA loss.

Overall, we conclude that translocation of the majority of MIM and matrix proteins was not affected in both patient fibroblasts and TIMM50 KD mice neurons, even after significant disruption of TIMM50 and the TIM23 core subunits. However, TIMM50 deficiency severely affected two major mitochondria complex systems, namely, the OXPHOS and MRP protein machineries.

## TIMM50 deficiency affects ATP production

The observed decrease in steady-state levels of OXPHOS subunits led us to examine oxygen consumption in TIMM50-deficient cells. For this purpose, we performed the Seahorse XF cell Mito Stress test (*Figure 5A*). Comparing oxygen consumption rates in the HC, P1, and P2 fibroblasts, and in the Sh2- and Scr control-transduced neuronal cells revealed significant impairment of both basal and maximal respiration rates and of OXPHOS-dependent energy production rate in the non-control cells (*Figure 5C and D*). Overall, these results suggest that TIMM50 deficiency severely affects the OXPHOS and MRP protein machineries and leads to OXPHOS-dependent ATP deficiency in both systems.

Moreover, as cells are able to meet their energetic requirements via glycolysis when the OXPHOS apparatus malfunctions (*Bhattacharya, 2023*), we measured the glycolytic capabilities of TIMM50-deficient cells by performing a Seahorse XF glycolysis stress test (*Figure 5B*). In both systems, the basal glycolysis level remained stable, in comparison to controls, suggesting that the cells did not make the metabolic switch so as to increasingly rely on the non-mitochondrial energy production pathway that is glycolysis (*Figure 5E and F*, left panel). Moreover, both glycolytic capacity and glycolytic reserves were significantly reduced, indicative of an impaired ability of both patient fibroblasts and neurons to switch their energetic emphasis to glycolysis when needed (*Figure 5E and F*, middle and right panels).

## TIMM50 KD impairs mitochondrial trafficking in neuronal cells

The transport of mitochondria within neuronal processes is crucial for cell survival (*Detmer and Chan, 2007*; *Johri and Beal, 2012*; *Wang et al., 2020*). Therefore, we investigated the effect of TIMM50 deficiency on mitochondrial trafficking in neuronal cells. To track individual mitochondria in neuronal processes, we used a transfection method, instead of transduction, that results in low expression efficiency in neurons (*Course et al., 2017*). This allowed us to visualize individual processes and track single mitochondria with minimal background noise. To visualize neuronal cells and their mitochondria, we co-transfected our cultures with a dsRed-mito-encoding plasmid, together with the KD or control plasmids. We then performed live cell imaging of individual neuronal processes and tracked the movement of individual mitochondria in these structures (for example, see *Figure 6A* and *Supplementary file 3*). The live imaging sets were converted into kymographs and calibrated in time and space, which allowed extraction of different trafficking parameters, such as distance of movement, speed, and percentage of moving mitochondria (*Figure 6B*).

Our results showed a twofold decrease in the percentage of mobile mitochondria in TIMM50-deficient neuronal cells, as compared to control cells (*Figure 6C*). Moreover, mobile mitochondria in

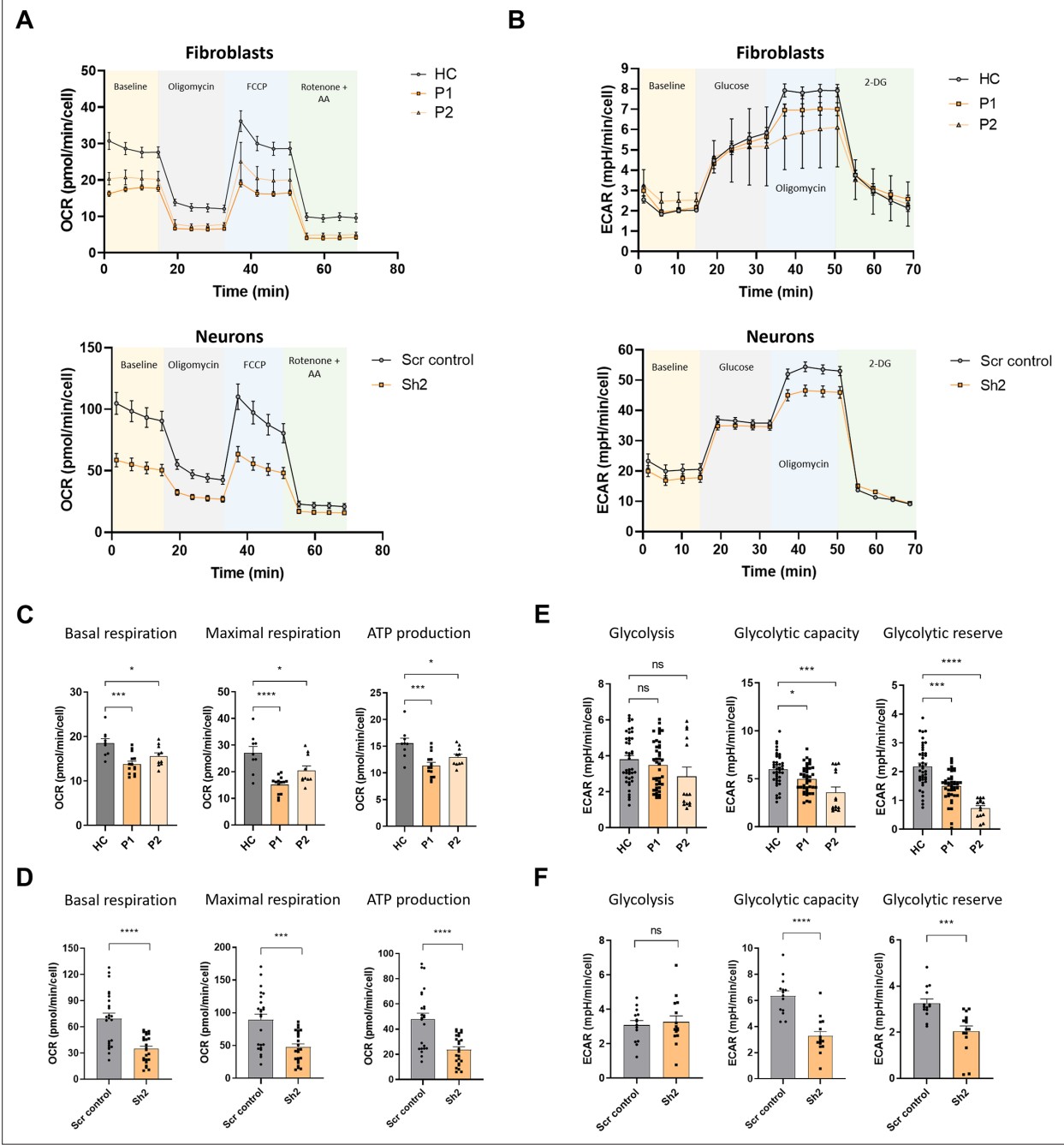

**Figure 5.** TIMM50 deficiency negatively impacts oxidative phosphorylation (OXPHOS) machinery and glycolysis functions. (**A**) A Seahorse XF cell mito-stress assay was used to measure mitochondrial oxygen consumption rates at basal levels and in response to the indicated effectors in fibroblasts (upper panel) and neuronal cells (lower panel). (**B**) A Seahorse XF cell glycolysis stress assay was used to measure glycolysis at basal levels and in response to the indicated effectors in fibroblasts (upper panel), and neuronal cells (lower panel). (**C**) Basal respiration, maximal respiration, and ATP-linked respiration were reduced in patient fibroblast cells compared to healthy control (HC) cells. Data are shown as means ± SEM, *p-value <0.05, ***p-value <0.001, ****p-value <0.0001, Ordinary one-way ANOVA. (**D**) Basal respiration, maximal respiration and ATP-linked respiration were reduced in TIMM50 KD neuronal cells compared to scrambled (Scr) control-transduced neuronal cells. Data are shown as means ± SEM, ***p-value <0.001, ****p-value <0.0001, unpaired Student's t-test. (**E**) Basal glycolysis remained similar, while glycolytic capacity and glycolytic reserves were reduced in patient fibroblast cells compared to HC cells. Data are shown as means ± SEM, *p-value <0.05, ***p-value <0.001, ****p-value <0.0001, Kruskal-Wallis test. (**F**) Basal glycolysis remained similar, while glycolytic capacity and glycolytic reserves were reduced in TIMM50 KD neuronal cells compared to Scr control-transduced neuronal cells. Data are shown as means ± SEM, ***p-value <0.001, ****p-value <0.0001, unpaired Student's t-test. For all the experiments shown in (**A–F**) n=4–6 biological repeats of three to six technical repeats each.

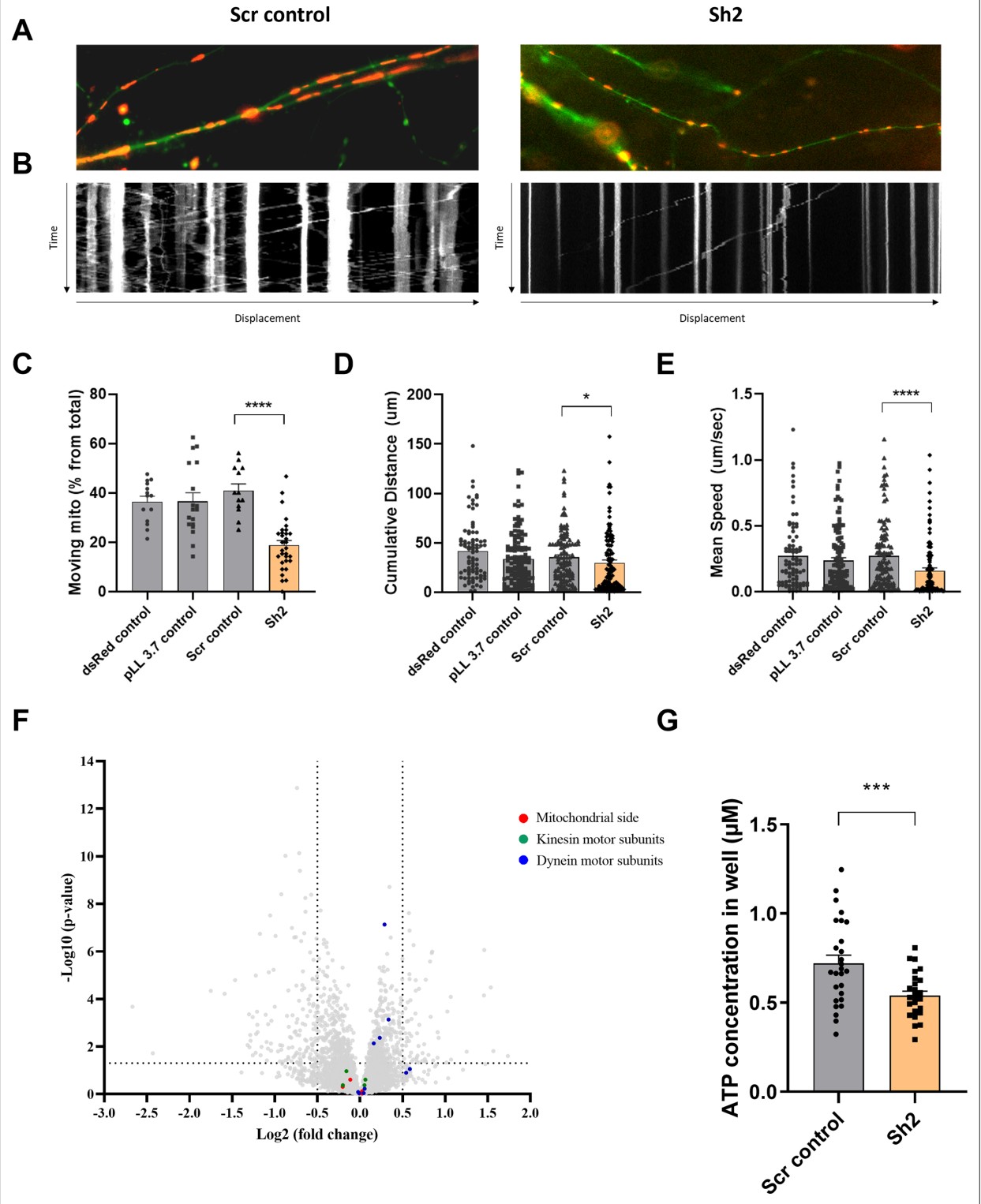

**Figure 6.** TIMM50 deficiency leads to defective mitochondrial trafficking along neuronal processes. (**A**) Representative images of neuronal processes in neuronal cultures that were co-transfected with a dsRed-mito plasmid and either a scrambled (Scr) control plasmid (left panel) or a TIMM50 knockdown (KD) plasmid (right panel). (**B**) Kymographs of the same processes in A, shows the displacement of mitochondria over time. The y-axis length is 5 min, and x-axis length is about 100 μm. (**C**) Lower percentage of moving mitochondria was observed in TIMM50 KD neuronal processes compared to controls. Each dot in the graph represents the percentage of moving mitochondria out of the total observed mitochondria in a single neurite. Data are shown as means ± SEM, n=13–31 neurites per condition, analyzed from three biological repeats, ****p-value <0.0001, Ordinary one-way ANOVA.

*Figure 6 continued on next page*

*Figure 6 continued*

(**D**) Decreased mitochondrial cumulative travelling distance was observed in TIMM50 KD neuronal cells compared to controls. Each dot in the graph represents a single moving mitochondrion. n=84–130 mitochondria per condition, analyzed from three biological repeats, *p-value <0.05, Kruskal-Wallis test. (**E**) Slower mitochondrial movement was observed in TIMM50 KD neuronal cells compared to controls. Each dot in the graph represents a single moving mitochondrion. n=84–130 mitochondria per condition, analyzed from three biological repeats, ****p-value <0.0001, Kruskal-Wallis test. (**F**) No change in mitochondrial trafficking proteins was observed in TIMM50 KD neuronal cells. The y-axis cut-off of >1.301 corresponds to –log (0.05) or p-value = 0.05, while the x-axis cut-off of <−0.5 and >0.5 corresponds to a ±1.414 fold change. Each dot in the graph represents a protein. Proteins depicted on the right side of the x-axis cut-off and above the y-axis cut-off were considered to be increased in amount, while proteins depicted on the left side of the x-axis cut-off and above the y-axis cut-off were considered to be decreased in amount. Statistical analysis was performed using Student's t-test and a p-value <0.05 was considered statistically significant. n=9 per group (three biological repeats in triplicate). Full list of differentially expressed proteins in neurons is found in **Source data 3**. (**G**) Lower cellular ATP levels were observed in TIMM50 KD neuronal cells compared to Scr control-transduced cells. Data are shown as mean ± SEM, n=27 quantified wells (each containing $5 \times 10^4$ cells) per condition, from three biological repeats with nine technical repeats each, ***p-value <0.001, unpaired Student's t-test.

TIMM50-deficient neuronal cells tend to cover less distance and travel at a lower average travelling speed (**Figure 6D and E**). This indicates that TIMM50 deficiency causes neuronal cell mitochondria to be more static, which could consequently lead to further energy deprivation in regions where mitochondria are needed but cannot be shipped.

Mitochondrial movement along neuronal processes is coordinated by a motor/adaptor complex. The motors kinesin and dynein use ATP to move organelles along microtubules. Mitochondria are assembled onto these motors via a mitochondrial outer membrane protein called Miro, a cytosolic adaptor called Milton (also known as TRAK1/2), and a few cytosolic accessory proteins (**Sheng, 2014**). Although TIM23 and TIMM50 are not directly involved in the biogenesis of any of these proteins, we, nonetheless, examined their expression levels following TIMM50 KD. Our proteomics data revealed no major changes in the levels of proteins involved in mitochondrial trafficking (**Figure 6F**), suggesting that the observed effect on neuronal cell mitochondrial trafficking is most likely indirect and resulting from the ATP deficiency. To examine this possibility, we measured cellular ATP levels in Scr control-transduced and TIMM50 KD neuronal cells. As expected from the impaired ATP production (**Figure 5**), we found that TIMM50 KD led to a significant reduction of about 25% in cellular ATP levels (**Figure 6G**).

## TIMM50 KD leads to excess neuronal activity and increased action potential frequency

All TIMM50 mutant patients studied thus far displayed severe neurological pathologies, which include epilepsy, developmental delay, and loss of movement abilities. Such abnormalities can be attributed to alternations in basic neuronal function. To assess whether such functions are altered upon TIMM50 KD in neuronal cells, we measured intrinsic neuronal excitability, as well as spontaneous neurotransmitter release, using the whole cell patch clamp technique.

Initially, we measured the spontaneous excitatory activity of the cells in the presence of tetrodotoxin (TTX) (representative traces are presented in **Figure 7—figure supplement 1A**). We quantified the average miniature excitatory post-synaptic current (mEPSC) amplitude, area, and frequency in each of the measured neuronal cells and found no significant differences between any of these measures in KD cells, as compared to controls (**Figure 7—figure supplement 1B**). Moreover, the relative frequency distribution of the amplitude measurements showed an even distribution pattern (**Figure 7—figure supplement 1C**). Examination of the cumulative distribution function confirmed that no significant differences in mEPSC amplitude distribution exist in the neuronal cells (**Figure 7—figure supplement 1D**).

We subsequently measured the minimal current required to induce an action potential by slowly increasing the stimulation current in a stepwise manner (**Figure 7A**). This allowed us to estimate the rheobase of the neuronal TIMM50 KD and control cells. We found that there were no significant differences in the current needed to trigger an action potential between the TIMM50 KD and Scr control-transduced cells (**Figure 7D**). We also assessed the characteristics of the first observed action potential in each measurement. Both the half-width and rate of fall of the first action potential were similar in TIMM50 KD and Scr control-transduced neurons (**Figure 7B, E and F**). However, TIMM50 KD cells displayed shorter action potential latency (**Figure 7G**) and a significant increase in the maximum

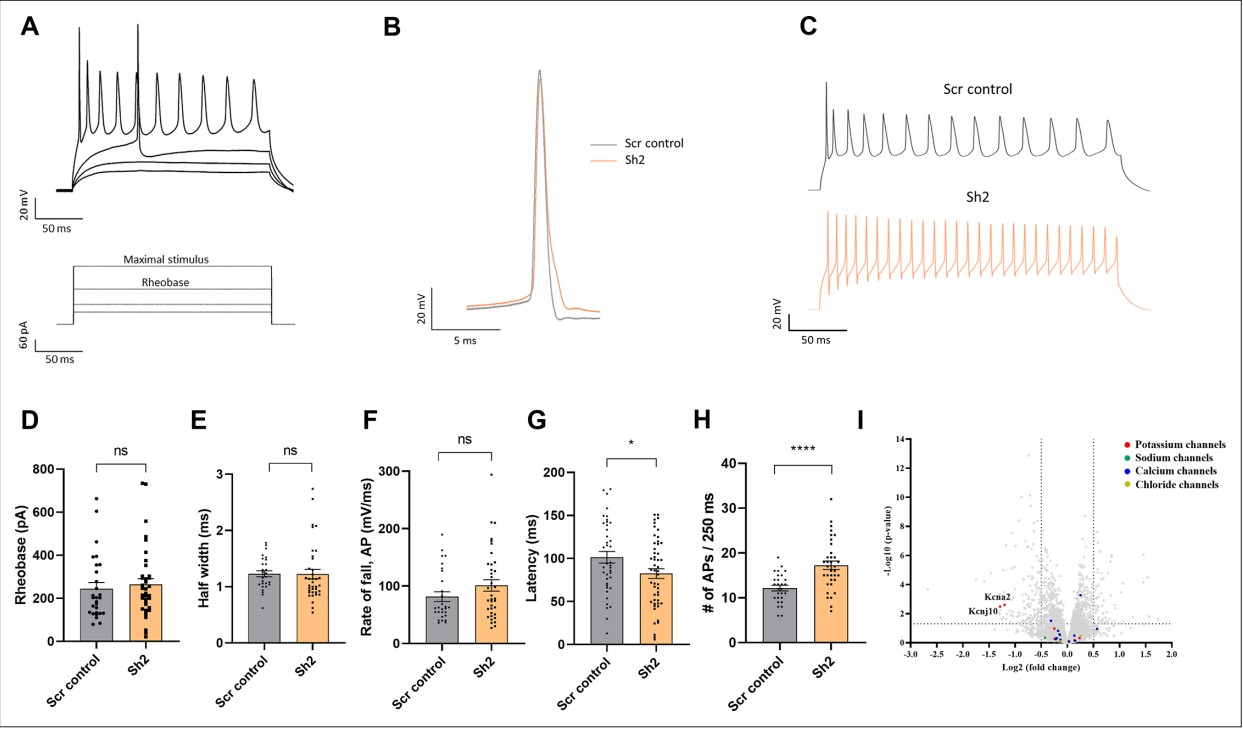

**Figure 7.** TIMM50 deficiency leads to a significant decrease in the levels of KCNA2 and KCNJ10 potassium channels and an increased electrical activity. (**A**) Example of voltage traces in response to increased depolarization of TIMM50 knockdown (KD) neurons. The bottom panel shows the protocol applied, while the top panel shows the typical response pattern that was measured. (**B**) Representative traces of a single action potential, received at rheobase level, from a TIMM50 KD neuron, as compared to Scr control-transduced neuron. (**C**) Representative traces of the maximal stimulus for each condition, showing the maximal amount of action potentials measured for each group. (**D**) No change in rheobase was observed in TIMM50 KD neuronal cells, as compared to scrambled (Scr) control-transduced cells. Rheobase was measured as the first current step that caused the firing of an action potential. Data are shown as means ± SEM, n=28/37 cells (for Scr control-/-Sh2-transduced cells, respectively) from three biological repeats, Mann-Whitney test. (**E**) No change in the action potential half-width was observed in TIMM50 KD neuronal cells, as compared to Scr control-transduced cells. Half-width was measured as the time between the rising and falling phases of the action potential, at the half-point between the tip of the peak and the bottom of the voltage rising curve. Data are shown as means ± SEM, n=28 / 37 cells (for Scr control-/ Sh2-transduced cells, respectively) from three biological repeats, Mann-Whitney test. (**F**) No change in the action potential rate of fall was observed in TIMM50 KD neuronal cells, as compared to Scr control-transduced cells. Rate of fall was measured as the time between the action potential peak and the baseline following the action potential, divided by the change in voltage between the same two points (ΔX/ΔY). Data are shown as means ± SEM, n=28/37 cells (for Scr control/ Sh2, respectively) from three biological repeats, Mann-Whitney test. (**G**) Decreased action potential latency was observed in TIMM50 KD neuronal cells compared to Scr control-transduced cells. Latency was measured as the time difference between the beginning of the pulse to the peak of the action potential. Each dot in the graph represents the average latency of the first five consecutive action potentials appearing after the rheobase level. Data are shown as means ± SEM, n=40/50 cells (for Scr control-/Sh2-transduced cells, respectively) from four biological repeats, *p-value <0.05, unpaired Student's t-test. (**H**) An increase in the maximal number of action potentials fired in a single stimulus was observed in TIMM50 KD neuronal cells, as compared to Scr control-transduced cells. Data are shown as means ± SEM, n=28/37 cells (for Scr control-/Sh2-transduced cells, respectively) from three biological repeats, ****p-value <0.0001, unpaired Student's t-test. (**I**) Amongst the detected ion channel proteins, a specific decrease in KCNA2 and KCNJ10 potassium channels was observed in TIMM50 KD neuronal cells. The y-axis cut-off of >1.301 corresponds to –log (0.05) or p-value = 0.05, while the x-axis cut off of <−0.5 and >0.5 corresponds to a ±1.414 fold change. Each dot in the graph represents a protein. Proteins depicted on the right side of the x-axis cut-off and above the y-axis cut-off were considered to be increased in amount, while proteins depicted on the left side of the x-axis cut-off and above the y-axis cut-off were considered to be decreased in amount. Statistical analysis was performed using Student's t-test and a p-value <0.05 was considered statistically significant. n=9 per group (three biological repeats in triplicate). Full list of differentially expressed proteins in neurons is found in *Source data 3*.

The online version of this article includes the following source data and figure supplement(s) for figure 7:

**Figure supplement 1.** Spontaneous excitatory activity recordings and the effect of α-dendrotoxin (α-DTX) on TIMM50 knockdown (KD) neuronal cells.

**Figure supplement 2.** Immunoblot confirmation of mass spectrometry-based proteomics findings for KCNA2.

**Figure supplement 2—source data 1.** PDF file containing original immunoblots for *Figure 7—figure supplement 2*, indicating the relevant bands and samples running order.

**Figure supplement 2—source data 2.** Original files for immunoblot analysis are displayed in *Figure 7—figure supplement 2*.

number of action potentials as compared to the Scr control-transduced cells (*Figure 7C and H*). Overall, these results suggest that TIMM50 KD causes the cells to fire more action potentials without decreasing the firing threshold, probably due to a faster recovery time between successive action potentials.

An increase in the firing frequencies of action potentials can be explained by a reduced presence of voltage-dependent potassium channels, which are known to control spike frequency (*D'Adamo et al., 2020*). Indeed, our proteomics results confirmed a decrease of about 2.5-fold in the levels of the KCNA2 and KCNJ10 potassium channels in the TIMM50-deficient neuronal cells, supporting the increase in their action potential frequency (*Figure 7I*). KCNA2 levels were also tested via immunoblot and confirmed to be dramatically decreased (*Figure 7—figure supplement 2*). To further test how KCNA2 reduction impacts cellular electrical activity, we used α-dendrotoxin (α-DTX), a known KCNA2 channel blocker (*Harvey, 2001*; *Glazebrook et al., 2002*), to mimic a reduction in KCNA2. The number of action potentials fired was measured before and after the application of 100 nM α-DTX to Scr control-transduced and TIMM50 KD neuronal cells (*Figure 7—figure supplement 1E*). As expected, the difference in the number of action potentials after and before α-DTX treatment increased significantly more in Scr control-transduced cells than in TIMM50 KD cells (*Figure 7—figure supplement 1F*), given how TIMM50 KD cells initially contain less KCNA2 channels than do the corresponding controls. These data indicate that a reduction in KCNA2 contributes to the observed increase in firing rate in TIMM50 KD neurons.

## Discussion

The TIMM50 protein is a pivotal member of the TIM23 complex that is suggested to participate in the import of nearly 60% of the mitochondrial proteome (*Schmidt et al., 2010*; *Pfanner, 2016*). Human *TIMM50* mutations lead to neurological effects, including mitochondrial epileptic encephalopathy, intellectual disability, seizure disorders like infantile spasms, and severe hypotonia accompanied by 3-methylglutaconic aciduria (*Serajee, 2015*; *Shahrour et al., 2017*; *Tort et al., 2019*; *Reyes et al., 2018*; *Mir et al., 2020*; *Moudi et al., 2022*). Despite being involved in the import of the majority of the mitochondrial proteome, little is known about the effects of TIMM50 deficiency on the entire mitochondrial proteome. Additionally, despite the fact that human TIMM50 mutations lead mainly to neurological symptoms, no study has yet addressed the effects of TIMM50 deficiency in brain cells. Therefore, in this study, we utilized two research models – TIMM50 mutant patient fibroblasts and TIMM50 KD primary mouse neuronal cultures, to study the impact of TIMM50 deficiency on the mitochondrial proteome and its impact on neurophysiology.

Interestingly, in both model systems, TIMM50 deficiency reduced the levels of TIM23 core subunits, yet did not alter steady-state levels of a majority of TIM23 substrates. These observations are similar to the recent analysis of patient-derived fibroblasts which demonstrated that *TIMM50* mutations lead to severe deficiency in the level of TIMM50 protein (*Reyes et al., 2018*; *Crameri et al., 2024*). Notably, this decrease in TIMM50 was accompanied by a decrease in the level of the other two core subunits, TIMM23 and TIMM17. However, unexpectedly, proteomics analysis in our study and that conducted by *Crameri et al., 2024* indicate that steady-state levels of most TIM23-dependent proteins are not affected despite a drastic decrease in the levels of the TIM23^CORE complex (*Crameri et al., 2024*). The most affected proteins constitute of intricate complexes, such as OXPHOS and MRP machineries. Thus, both these studies might indicate that even reduced levels of the TIM23^CORE components are sufficient for maintaining the steady-state levels of most presequence-containing substrates. This is surprising, as normal TIM23 complex levels are suggested to be indispensable for the translocation of presequence-containing mitochondrial proteins (*Mokranjac et al., 2003*; *Yamamoto et al., 2002*; *Vögtle et al., 2009*; *Pfanner and Geissler, 2001*; *Chacinska et al., 2005*). Even more surprising was that the amounts of some TIM23 substrates related to intricate metabolic and maintenance activities (e.g. ALDH2, GRSF-1, OAT, etc.) were increased.

These observations can be explained by several plausible mechanisms: it is possible that unlike what occurs in yeast, normal levels of mammalian TIMM50 and TIM23 complexes are mainly essential for maintaining the steady-state levels of intricate complexes/assemblies. Another explanation for this scenario is that the normal quantities of unaffected matrix proteins might be low, and hence, even ~10–20% of functional TIMM50 protein might be sufficient to maintain their steady-state levels. Alternatively, the presequence of such proteins might contain mitochondrial targeting signals (MTS)

that receive priority over other presequence-containing precursor proteins, thus enabling their translocation even in the presence of very few functional TIM23 complexes. However, further experiments examining these possibilities are needed to understand the compromised TIM23-mediated protein import.

As stated earlier, the loss of TIMM50 led to a significant reduction in the steady-state levels of other TIM23 core subunits (i.e. TIMM23 and TIMM17A/B). Notably, TIMM23 and TIMM17A/B are thought to be imported by the TIM22 complex (*Káldi et al., 1998*). However, we observed that the steady-state levels of TIM22 complex subunits had not been affected by TIMM50 deficiency (Files 1-3*Source data 1*; *Source data 2*; *Source data 3*). This indicates a hitherto unknown relation between steady state levels of TIMM50 and other TIM23 core subunits, which might affect the complex assembly process. Various structural and biochemical studies have attempted to elucidate the intricate structure of the TIM23 complex and the complicated interactions between its different subunits (*Sim et al., 2023*; *Fielden et al., 2023*; *Gevorkyan-Airapetov et al., 2009*), however, little is presently known of the dynamic assembly processes of the complex. The fact that TIMM50 KD leads to a major reduction in the levels of TIMM23 and TIMM17A/B, despite a lack of direct dependency on the import of these proteins on TIMM50, and does not affect the levels of other TIM23 complex subunits, could provide a basis for future studies examining the assembly process of import complexes.

Our proteomics data and Seahorse XF analysis, paired with cellular ATP measurements, indicated that lower ATP levels were present in TIMM50-deficient cells. Specifically in the case of neurons, such ATP deficiency could be responsible for the negative impact seen on mitochondrial trafficking in neuronal cell processes, which could contribute to the various neurodegenerative phenotypes linked to the TIMM50 disease. Additionally, the detected increase in action potential firing rate in TIMM50-deficient neurons can explain the presence of epileptic seizures, a hallmark of all TIMM50 patients studied thus far. A plausible explanation for the increased action potential firing rate could be the 2.5-fold reduction in the levels of the KCNA2 ($K_v$1.2) and KCNJ10 ($K_{ir}$4.1) voltage-dependent potassium channels, as revealed by our proteomics and immunoblot analysis. The KCNA2 ($K_v$1.2) channel is a slowly inactivating channel that regulates neuronal excitability and firing rate (*Shen et al., 2004*). In peripheral sensory neurons, $K_v$1.2 helps to determine spike frequency, while in neurons of the medial trapezoid body, the Kv1.1, Kv1.2, and Kv1.6 subunits are important regulators of repetitive spiking (*Glazebrook et al., 2002*; *Dodson et al., 2002*). While the inactivation of several of the potassium channels like $K_v$1.2, $K_v$1.1, and Kcnj10 ($K_{ir}$4.1) are important for neuronal excitability, action potential width, and firing properties, in general, mutations in the genes encoding these channels that alter their inactivation are known to lead to temporal lobe epilepsy (*Shen et al., 2004*; *Kaczmarek, 2006*; *Schulte et al., 2006*). In addition, blocking the activation of the $K_v$1.2 by α-DTX had no effect on the resting membrane potential and only small effects on the amplitude and duration of the action potential (*Glazebrook et al., 2002*), similar to what we observed in TIMM50 KD neurons, that showed a reduction in $K_v$1.2 levels. Furthermore, blocking $K_v$1.2 activation by α-DTX increased the frequency of action potentials in visceral sensory neurons (*Glazebrook et al., 2002*), which, again, agrees with our observation that α-DTX application increased the firing rate in Scr control-transduced neurons, while hardly affecting the firing rate of TIMM50 KD neurons as they express lower levels of $K_v$1.2. Hence, similar changes in the neuronal firing rate, as observed in our TIMM50 KD neurons, might occur in TIMM50 patients, and could lead to the epileptic phenotype seen in these patients. However, more studies are needed to verify this hypothesis.

In summary, our results suggest that even low levels of TIMM50 and TIM23[CORE] components suffice in maintaining the majority of the mitochondrial matrix and inner membrane proteome. Nevertheless, a reduction in TIMM50 levels leads to a decrease of many OXPHOS and MRP complex subunits, which indicates that normal TIMM50 levels might be mainly essential for maintaining the steady state levels and assembly of intricate complex proteins. The consequently reduced cellular ATP levels and the detected mitochondrial abnormalities in neurons provide a plausible link between the TIMM50 mutation and the observed developmental defects in the patients. Moreover, the increased electrical activity resulting from decreased steady-state levels of KCNA2 and KCNJ10 potassium channels plausibly links the TIMM50 mutation to the epileptic phenotype of patients, thus, providing a new direction for therapeutic efforts.

## Materials and methods

### Generation of primary human fibroblasts

Four mm punch biopsy samples were obtained from two TIMM50 patients Patient 1 (P1) and Patient 2 (P2) carrying the mutation c.446C>T; p.Thr149Met and a normal family member that served as a healthy control (HC). Primary fibroblast cells were generated using standard procedures (*Vangipuram et al., 2013*). In brief, each biopsy sample was cut into 12–15 pieces, and 2–3 pieces were placed in six-well plate wells containing complete Dulbecco's modified Eagle's medium (DMEM; DMEM, 20% fetal bovine serum (FBS), 1% sodium pyruvate, 1% penicillin-streptomycin) and previously coated with 0.1% gelatin. The fibroblasts were grown for 2–3 wk and passaged into 10 cm plates. Cells obtained from the first three passages were frozen and stored for further use. Following the generation of the primary fibroblast cells, genomic DNA purification and sequencing were performed to verify the presence of the mutation. Sequencing primers are found in *Supplementary file 1*.

### Generation of TIMM50 KD mice primary cortical neuronal cultures

Mouse primary cortical neurons were harvested from $P_0/P_1$ pups and cultured using a previously described procedure (*Lavi et al., 2014*). Plates were pre-coated with Matrigel (Corning, 354234) (diluted 1:1000 in Hank's balanced salt solution (Sartorius, 02-018-1A) with 10 mM HEPES, pH 7.4 (Fisher bioreagents, BP310-500)). For TIMM50 knockdown, three targeting shRNA sequences (Sh1, Sh2 and Sh3) and a scrambled (Scr) control sequence were designed. All shRNA sequences were cloned into the third-generation lentiviral vector pLL3.7 for expression under the control of the U6 promotor. The same plasmid also encoded EGFP under control of the hSyn promotor, which allowed us to visualize and differentiate neuronal cells from other cells in the culture. To produce lentiviral particles, HEK293T/17 (ATCC, CRL-11268, cells tested negative for mycoplasma and were authenticated by STR profiling) cells were co-transfected with each of the designed shRNA vectors, together with the lentiviral helper constructs pMDLg-pRRE, pRSV-REV, and CMV-VSVG, via calcium phosphate transfection. To generate TIMM50 KD neurons, the neuronal cultures were transduced on 4 d in vitro (DIV) with the generated lentiviruses and grown until 18 DIV. Immunoblotting with anti-TIMM50 antibodies were used to assess KD efficiency. shRNA oligo sequences and pLL3.7 sequencing primers are found in *Supplementary file 1*.

### Immunoblotting

Fibroblasts were grown to ~90% confluency on 10 cm cell culture plates, harvested using trypsin, washed twice with PBS, and then lysed using 50 µl of solubilization buffer 50 mM Na-HEPES, pH 7.4, 150 mM NaCl, 1.5 mM $MgCl_2$, 10% glycerol, 1% Triton X-100, 1 mM EDTA, 1 mM EGTA supplemented with 400 µM of PMSF and 1:1000 dilution of protease inhibitor cocktail (GenDEPOT, P3200-020). Neurons (~1 × $10^6$ cells) were cultured in the wells of a six-well plate, transduced on 4 DIV, grown until 18 DIV, and lysed by adding 50 µl of solubilization buffer to each plate well, followed by scraping with a cell scraper. The protein concentration of both lyzed cultures was measured using Bradford reagent (BioRad, 500–0006), and appropriate protein amounts (20–100 µg) were loaded onto home-made polyacrylamide gels (12/14/16%) and separated using sodium dodecyl sulfate-polyacrylamide gel electrophoresis. The separated proteins were then transferred to a PVDF membrane (Millipore, IPVH00010) and immunodetection was carried out using antibodies against the target proteins. The list of antibodies used can be found in *Supplementary file 2*. For fibroblasts, actin or GAPDH served as a loading control. For neurons, tubulin was used as a loading control. ImageJ was used for densitometry analysis of protein expression levels. At least three biological repeats were performed for each immunoblotted protein (original blots for every biological repeat performed in every immunoblot analysis are displayed in the source data files for the relevant figures).

### Quantitative protein assessment and analysis

Label-free quantitative mass spectrometry was performed based on a published procedure (*Seyfried et al., 2017*). Spectra were searched against the Uniprot/Swiss-Prot mouse database (17,041 target sequences) for neuronal cells or the Uniprot/Swiss-Prot human database (20,379 target sequences) for fibroblast cells using the Andromeda search engine integrated into MaxQuant. Methionine oxidation (+15.9949 Da), asparagine and glutamine deamidation (+0.9840 Da), and protein N-terminal acetylation (+42.0106 Da) were variable modifications (up to five allowed per

peptide), while cysteine was assigned a fixed carbamidomethyl modification (+57.0215 Da). Trypsin-cleaved peptides with up to two missed cleavages were considered in the database search. A precursor mass tolerance of ±20 ppm was applied prior to mass accuracy calibration and ±4.5 ppm after internal MaxQuant calibration. Other search settings included a maximum peptide mass of 6000 Da, a minimum peptide length of 6 residues, 0.05 Da tolerance for Orbitrap, and 0.6 Da tolerance for ion trap MS/MS scans. The false discovery rates for peptide spectral matches, proteins, and site decoy fractions were all set to 1 percent. Quantification settings were as follows: Re-quantification in a second peak finding attempt after protein identification; matched MS1 peaks between runs; a 0.7 min retention time match window after an alignment function was found with a 20 min RT search space. Protein quantitation was performed using summed peptide intensities provided by MaxQuant. The quantitation method only considered razor plus unique peptides for protein-level quantitation.

Data were obtained from three biological repeats, each involving three technical repeats (i.e. nine samples in total) for every cell type (neurons: Sh2 and Scr control; fibroblasts: P1, P2, and HC). Statistical analysis was performed using Perseus software. Protein levels were considered to be increased or decreased in TIMM50-deficient cells if they were significantly different (p-value <0.05) and had a fold-change of at least 1.414, relative to what was measured in control cells. Mitochondrial protein classification was performed manually by comparing the obtained data with the MitoCarta3.0 human and mouse databases (*Rath et al., 2021*). Go-term analysis was performed using the database for annotation, visualization, and integrated discovery (DAVID) (*Sherman et al., 2022*).

## Seahorse XF mito-stress and glycolysis stress tests

Fibroblasts were plated in Seahorse XF 96-well plates (Agilent, 103775–100) at 20–30% confluency, with experiments being carried out at ~90% confluency. For neurons, ~1 × 10^5 cells were plated in each well of Seahorse XF 96-well plates, transduced with the Scr control or Sh2 constructs on 4 DIV and the experiment was carried out on 18 DIV. One-two hours prior to the experiment, the fibroblasts or neuronal cultures were washed and the medium was replaced with Seahorse XF DMEM, pH 7.4 (Agilent, 103575–100).

For the mito-stress test, the medium was supplemented with 1 mM sodium pyruvate (Sigma, S8636-100ML), 10 mM glucose (Merck, 1.08337.1000), and 2 mM glutamine (Biological Industries, 03-020-1B). The plates were loaded with oligomycin (Sigma-Aldrich, O4876), carbonyl cyanide-p-trifluoromethoxyphenylhydrazone (FCCP, Sigma-Aldrich, C2920), and rotenone (Sigma-Aldrich, R8875) together with antimycin A (Sigma-Aldrich, A8674), at final well concentrations of 1, 2, 0.5, and 0.5 µM, respectively. For the glycolysis stress test, the medium was only supplemented with glutamine. The plates were loaded with glucose, oligomycin, and 2-deoxy-D-glucose (2-DG, Sigma-Aldrich, D8375-1G) at final well concentrations of 10 mM, 1 µM, and 50 mM, respectively. Plates were then loaded into the Seahorse XFe96 Extracellular Flux Analyzer and the experiments were carried out using the manufacturer's protocol. To normalize the oxygen consumption rate (OCR) or extracellular acidification rate (ECAR) values, fibroblasts were dyed with SynaptoGreen (Biotium, 70022) immediately at the end of each experiment and fluorescence levels were measured using a plate reader (BioTek Synergy HTX). In the case of neurons, the cells were dyed with DRAQ5 (BioLegend, 424101) immediately at the end of each experiment, and imaged on an Incucyte SX5 live cell imaging and analysis system. Dyed nuclei were counted using the ImageJ particle analysis function. Three to six biological repeats, each involving three to six technical repeats, were performed for each experimental and control group.

## Mitochondrial DNA content

Total DNA was isolated form near-confluent fibroblasts or 18 DIV transduced neuronal cultures using a GeneElute Mammalian Genomic DNA Miniprep kit (Sigma, G1N350-1KT). Quantitative real-time PCR (StepOnePlus Real-Time PCR System) was then performed on each sample in the presence of SYBR green (PCR Biosystems, PB20.16–05). Expression levels were determined using the comparative cycle threshold ($2^{-\Delta\Delta Ct}$) method, with the hypoxanthine guanine phosphoribosyl transferase (HPRT)-encoding gene serving as housekeeping gene. Primer sequences used are listed in *Supplementary file 1*.

## Mitochondrial trafficking

Neuronal cultures were plated in a similar manner as described for membrane potential measurements. Neuronal cultures (4 DIV) were co-transfected with the TIMM50 KD or control plasmids, as well as a mito-dsRed-expressing plasmid (*Lindenboim et al., 2013*), using 0.6 µg of each plasmid and 0.6 µl of Lipofectamine 2000 (Invitrogen, 11668–027). A transfection rate of 2–5% was seen the next day. This sparse transfection allowed for visualization of single neurons and their mitochondria, given the dramatically reduced background that allowed for tracking of mitochondrial movement in individual neurites. Following transfection, 10 DIV cultures were live-imaged with an iMIC inverted microscope equipped with a Polychrome V system (TILL photonics) and an ANDOR iXon DU 888D EMCCD camera (Andor, Belfast, Northern Ireland). Cells were imaged using a 60 x oil immersion objective (Olympus), under temperature and $CO_2$ control. Fields of view containing neurites stretching for at least 50 µm and not more than 100 µm from the cell body were imaged for 5 min, with a 3 s interval between each image. Individual neurites in each image set were selected using the segmented line tool in ImageJ. The images were then calibrated in time and space and turned into kymographs using the KymoToolBox ImageJ plugin (*Zala et al., 2013*). Each individual mitochondrion present in the neurite was then manually tracked on the kymograph using the segmented line tool to extract various parameters. Mitochondria were defined as static if they moved at a speed lower than 0.02 µm/s.

## Determination of cellular ATP levels

Neuronal cells were plated into a 96-well cell culture plate at a density of $5 \times 10^4$ cells per well. Cells were transduced with the Sh2 or Scr control constructs on 4 DIV and cellular ATP measurements were performed on 18 DIV using a Luminescent ATP Detection Assay Kit (Abcam, ab113849). To block luminescence signal contamination from adjacent wells, the lysis step of the assay was performed in the clear cell culture plate and the lysates were transferred to a white, flat-bottom 96-well plate. Luminescence signals were read using the GloMax Navigator System.

## Intrinsic neuronal excitability and spontaneous activity

Neurons were plated at a density of $1.5 \times 10^5$ cells/well of a 12-well plate and transduced on 4 DIV, with experiments being carried out on 16–20 DIV. Conventional whole-cell recordings were performed with borosilicate thin wall glass capillaries (World Precision Instruments, TW150-3) with an input resistance of 4–5 MΩ. Series resistance ranged from 8 to 25 MΩ. An EPC-9 patch clamp amplifier was used in conjunction with PatchMaster software (HEKA Electronik, Lambrecht, Germany). The external solution consisted of 140 mM NaCl, 3 mM KCl, 2 mM $CaCl_2$, 1 mM $MgCl_2$, 10 mM HEPES, supplemented with 2 mg/ml glucose, pH 7.4, and osmolarity adjusted to 305 mOsm. The internal solution consisted of 110 mM K gluconate, 10 mM KCl, 2 mM $MgCl_2$, 10 mM HEPES, 10 mM glucose, 10 mM Na creatine phosphate, and 10 mM EGTA, pH 7.4. Osmolarity adjusted to 285 mOsm.

To measure the rheobase and assess repetitive and maximal firing, long (250 ms) depolarization square current steps of varying intensity (from –100–600 pA, at 2 pA intervals) were applied. Signals were filtered at 2 kHz and sampled at 5 kHz. For α-dendrotoxin (α-DTX) measurements, long (250 ms) depolarizing square current steps of varying intensity (From 0–500 pA, at 100 pA intervals) were applied to the cells, before and after application of an external solution containing 100 nM α-DTX (Alomone Labs, D-350). Clear traces in the range of 100–300 pA were chosen in the before-α-DTX measurements and compared to the respective traces in the after-α-DTX measurements. To measure spontaneous activity, an external solution containing 1 µM tetrodotoxin (TTX; Alomone Labs, T550_1 mg) was perfused onto the cells, followed by a high current injection to assure that no action potentials were being generated. The mode was then switched to a voltage clamp, the holding voltage was adjusted to –60 mV and mEPSCs were recorded for 2 min (at repeating 10 s intervals). The data were analyzed with Igor Pro software (Wavemetrics, Lake Oswego, OR). The TaroTools procedure set (for Igor Pro) was used to analyze spontaneous activity measurements.

## Acknowledgements

The authors thank Professor Bernard Attali, Dr. Celeste Weiss Katz, Dr. Natalia Borovok, and Dr. Amit Kessel for their valuable input and guidance, and to the undergrad project students Afek Moravia and Avia Tamir for their technical support. This work was supported by the Emory University Emory

Integrated Proteomics Core Facility (RRID:SCR_023530). AA is incumbent of The Louise and Nahum Barag Chair in Molecular Genetics of Cancer. UA is the incumbent of the Michael Gluck Chair in Neuropharmacology and ALS Research.Work in the Azem lab was supported by grants #1389/18 and #1057/22 from the Israel Science Foundation, and an Emory University and Tel Aviv University Collaborative Research Grant. This research was also supported by the Ministry of Innovation, Science & Technology, Israel (1001576154), Israel Science Foundation (ISF grant: 2141/20), BrightFocus grant (A2022029S), NIH grant 1R21AG074846-01A1, and the Michael J Fox Foundation (MJFF-022407) (to UA).

## Additional information

### Funding

| Funder | Grant reference number | Author |
|---|---|---|
| Israel Science Foundation | 1389/18 | Abdussalam Azem |
| Israel Science Foundation | 1057/22 | Abdussalam Azem |
| Ministry of Innovation, Science and Technology | 1001576154 | Uri Ashery |
| Israel Science Foundation | 2141/20 | Uri Ashery |
| BrightFocus Foundation | A2022029S | Uri Ashery |
| Michael J. Fox Foundation for Parkinson's Research | MJFF-022407 | Uri Ashery |
| NIH Blueprint for Neuroscience Research | 1R21AG074846-01A1 | Uri Ashery |

The funders had no role in study design, data collection and interpretation, or the decision to submit the work for publication.

### Author contributions

Eyal Paz, Sahil Jain, Data curation, Formal analysis, Investigation, Writing - original draft; Irit Gottfried, Resources, Validation, Writing - review and editing; Orna Staretz-Chacham, Muhammad Mahajnah, Resources; Pritha Bagchi, Investigation, Writing - review and editing; Nicholas T Seyfried, Supervision, Methodology; Uri Ashery, Abdussalam Azem, Conceptualization, Supervision, Funding acquisition, Validation, Methodology, Writing - review and editing

### Author ORCIDs

Eyal Paz ⓘ https://orcid.org/0009-0009-3674-956X
Sahil Jain ⓘ https://orcid.org/0000-0003-3799-2421
Abdussalam Azem ⓘ https://orcid.org/0000-0002-2288-2986

### Ethics

This study was conducted under Helsinki license (202227695) and was approved by the university ethics committee (0005773-6). Participants gave written informed consent.
Our laboratories adhere to the latest guidelines (ARRIVE) on good clinical practice for animal studies. Animals were approved by the Tel Aviv University Animal Care and Use Committee (Approval Number TAU - LS - IL - 2205 - 154 - 3).

Reviewer #1 (Public review): https://doi.org/10.7554/eLife.99914.3.sa1
Reviewer #2 (Public review): https://doi.org/10.7554/eLife.99914.3.sa2
Author response https://doi.org/10.7554/eLife.99914.3.sa3

## Additional files

### Supplementary files
- Supplementary file 1. Oligonucleotides were used in this study.
- Supplementary file 2. Reagents and tools used in this study.
- Supplementary file 3. Representative mitochondrial trafficking movies. Scr control neurons and TIMM50 knockdown (KD) neurons co-transfected with the corresponding control/KD plasmid and mito-DsRed plasmid. The cells were imaged for 5 min with a 3 s interval between each image. The videos displayed are sped up by ~15x.
- MDAR checklist
- Source data 1. Healthy control (HC) vs P1 all proteins fold change.
- Source data 2. Healthy control (HC) vs P2 all proteins fold change.
- Source data 3. Scrambled (Scr) vs Sh2 all proteins fold change.

### Data availability

Proteomics data generated are available on Synapse, and all data related to gels and westerns are available in supplementary materials.

The following dataset was generated:

| Author(s) | Year | Dataset title | Dataset URL | Database and Identifier |
|---|---|---|---|---|
| Paz E, Jain S, Gottfried I, Staretz-Chacham O, Mahajnah M, Bagchi P, Seyfried NT, Ashery U, Azem A | 2024 | Fibroblasts and Neurons proteomics datasets | https://www.synapse.org/Synapse: syn64028721 | Synapse, Syn64025159 |

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
