## [Editor Report · eLife Assessment]

This **important** study presents interesting results aimed at explaining the effects of a human mutation on the mitochondrial import protein TIMM50 on mitochondrial function and neuronal excitability. While the evidence supporting the conclusions is **convincing**, the mechanisms driving changes in the levels of certain proteins within and outside the mitochondria (such as certain ion channels) remain unexplained. This paper will be of interest to scientists in the mitochondria field.

---

## [Referee Report · Reviewer #1 (Public review)]

Mitochondria are essential organelles consisting in mammalian cells of about 1500 different proteins. Most of those are synthesized in the cytosol as precursor proteins, imported into mitochondria, and sorted into one of the four sub-mitochondrial compartments. The TIM23 complex, which is embedded in the mitochondrial inner membrane, facilitates the import of proteins that harbor Mitochondrial Targeting Sequence (MTS) at their N-terminus. Such proteins are sorted mainly to the mitochondrial matrix while some sub-groups are destined also to the inner membrane or the intermembrane space. TIMM50 (Tim50 in yeast) is an essential component of the TIM23 complex and mutations in this protein were reported to cause several diseases.

Summary:

In the current study, the authors analyzed the impact of TIMM50 mutations on the mitochondrial proteome in both patients' cells and mouse neurons. They provide compelling evidence for several surprising and highly interesting observations: (i) TIMM50 mutations affect the steady-state levels of only a portion of the putative TIMM50 substrates, (ii) such mutations result in increased electrical activity in mice neurons and in reduced levels of some potassium ion channels in the plasma membrane. These findings shed new light on mitochondrial biogenesis in mammalian cells and hint at an unexpected link between mitochondria and ion channels at the plasma membrane.

Strengths:

The authors used both cells from patients and neurons from mice to investigate the impact of mutations in TIMM50 on mitochondrial proteome and function.

Comments on revisions:

The authors addressed all my concerns regarding the original submission.

---

## [Referee Report · Reviewer #2 (Public review)]

Summary:

Mitochondria import hundreds of precursor proteins from the cytosol. The TOM and TIM23 complexes facilitate the import on the matrix-targeting pathway of mitochondria. In yeast, Tim50 is a critical and essential subunit of the TIM23 complex that mediates the transition of precursors from the outer to the inner membrane. The human Tim50 homolog TIMM50 is highly similar in structure and a comparable function of Tim50 and TIMM50 was proven by several biochemical and genetic studies in the past.

In this study, the authors characterize human cells which express lower levels or mutated versions of TIMM50. They found that in these TIMM50-depletion cells, the levels of other TIM23 core subunits are also diminished but many mitochondrial proteins are unaffected. Moreover, they observed alterations in the electrical activity and the levels of potassium channels in neuronal cells of TIMM50-deficient mice. They propose that these changes explain the pathology of patients who often suffer from epilepsy.

Strengths:

The paper is written by experts in the field, and it is very clear. The experiments are of high quality and sufficiently well-controlled. The study is interesting for a broad readership.

Weaknesses:

The authors show that even upon low levels of Tim50, mitochondrial proteins are not considerably depleted. However, it remains somewhat unclear why this is. TIMM50 and the TIM23 complex might not be rate-limiting for the biogenesis of mitochondrial proteins. Alternatively, the import defect is compensated indirectly, for example by a reduced growth of cells. It will be interesting to study the physiological consequences of TIMM50-depletion in more depth in the future.

---

## [Author Response]

The following is the authors’ response to the original reviews.

We thank the reviewers and editor for their positive view and constructive valuable comments on the manuscript. Following we address the suggestions of the reviewers.

**Reviewer #1 (Public Review):**
(1) It will be interesting to monitor the levels of another MIM insertase namely, OXA1. This will help to understand whether some of the observed changes in levels of OXPHOS subunits are related to alterations in the amounts of this insertase.

OXA1 was not detected in the untargeted mass spectrometry analysis, most likely due to the fact that it is a polytopic membrane protein, spanning the membrane five times (1,2). Consequently, we measured OXA1 levels with immunoblotting, comparing patient fibroblast cells to the HC. No significant change in OXA1 steady state levels was observed.

These results are now displayed (Fig. S3B and C) and discussed in the revised manuscript.

Figure 3: How do the authors explain that although TIMM17 and TIMM23 were found to be significantly reduced by Western analysis they were not detected as such by the Mass Spec. method?

The untargeted mass spectrometry in the current study failed to detect the presence of TIMM17 for both, patient fibroblasts and mice neurons, while TIMM23 was detected only for mice neurons and a decrease was observed for this protein but was not significant. This is most likely due to the fact that TIMM17 and TIMM23 are both polytopic membrane proteins, spanning the membrane four times, which makes it difficult to extract them in quantities suitable for MS detection (2,3).

(2) How do the authors explain the higher levels of some proteins in the TIMM50 mutated cells?

The levels of fully functional TIM23 complex are deceased in patients' fibroblasts. Therefore, the mechanism by which the steady state level of some TIM23 substrate proteins is increased, can only be explained relying on events that occur outside the mitochondria. This could include increase in transcription, translation or post translation modifications, all of which may increase their steady state level albite the decrease in the steady state level of the import complex.

(3) Can the authors elaborate on why mutated cells are impaired in their ability to switch their energetic emphasis to glycolysis when needed?

Cellular regulation of the metabolic switch to glycolysis occurs via two known pathways: (1) Activation of AMP-activated protein kinase (AMPK) by increased levels of AMP/ADP (4). (2) Inhibition of pyruvate dehydrogenase (PDH) complexes by pyruvate dehydrogenase kinases (PDK) (5). Therefore, changes in the steady state levels of any of these regulators could push the cells towards anaerobic energy production, when needed. In our model systems, we did not observe changes in any of the AMPK, PDH or PDK subunits that were detected in our untargeted mass spectrometry analysis (see volcano plots below, no PDK subunits were detected in patient fibroblasts). Although this doesn’t directly explain why the cells have an impaired ability to switch their energetic emphasis, it does possibly explain why the switch did not occur de facto.

**Reviewer #2 (Public Review):**
(1) The authors claim in the abstract, the introduction, and the discussion that TIMM50 and the TIM23 translocase might not be relevant for mitochondrial protein import in mammals. This is misleading and certainly wrong!!!

Indeed, it was not in our intention to claim that the TIM23 complex might not be relevant. We have now rewritten the relevant parts to convey the correct message:

Abstract –

Line 25 - “Strikingly, TIMM50 deficiency had no impact on the steady state levels of most of its putative substrates, suggesting that even low levels of a functional TIM23 complex are sufficient to maintain the majority of complex-dependent mitochondrial proteome.”

Introduction –

Line 87 - Surprisingly, functional and physiological analysis points to the possibility that low levels of TIM23 complex core subunits (TIMM50, TIMM17 and TIMM23) are sufficient for maintaining steady-state levels of most presequence-containing proteins. However, the reduced TIM23CORE component levels do affect some critical mitochondrial properties and neuronal activity.

Discussion –

Line 339 – “…surprising, as normal TIM23 complex levels are suggested to be indispensable for the translocation of presequence-containing mitochondrial proteins…”

Line 344 – “…it is possible that unlike what occurs in yeast, normal levels of mammalian TIMM50 and TIM23 complex are mainly essential for maintaining the steady state levels of intricate complexes/assemblies.”

Line 396 – “In summary, our results suggest that even low levels of TIMM50 and TIM23CORE components suffice in maintaining the majority of mitochondrial matrix and inner membrane proteome. Nevertheless, reductions in TIMM50 levels led to a decrease of many OXPHOS and MRP complex subunits, which indicates that normal TIMM50 levels might be mainly essential for maintaining the steady state levels and assembly of intricate complex proteins.”

**Reviewer #1 (Recommendations For The Authors):**
(1) Lines 25-26: The authors write "Strikingly, TIMM50 deficiency had no impact on the steady state levels of most of its substrates". Since the current data challenges the definition of some proteins as substrates of TIMM50, I suggest using the term "putative substrates".

Changed as suggested

(2) Line 27: It is not clear whether the wording "general import role of TIM23" it refers to the TIM23 protein or the TIM23 complex. This should be clarified.

Clarified. It now states "TIM23 complex".

(3) Line 72: should be "and plays".

Changed as suggested.

(4) It will be helpful to include in Figure 1 a small scheme of TIMM50 and to indicate in which domain the T252M mutation is located.

We predicted the AlphaFold human TIMM50 structure and indicated the mutation site and the different TIMM50 domains. The structure is included in Fig. 1A.

(5) I suggest labelling the "Y" axis in Fig. 1B as "Protein level (% of control)".

Changed as suggested in Fig. 1C (previously Fig. 1B) and in Fig. 2C.

(6) Line 179: since the authors tested here only about 10 mitochondrial proteins (out of 1500), I think that the word "many" should be replaced by "several representative" resulting in "steady state levels of several representative mitochondrial proteins".

Changed as requested.

(7) Line 208: correct typo.

Typo was corrected.

(8) Figure 4 is partially redundant as its data is part of Figure 3. The authors can consider combining these two figures. Accordingly, large parts of the legend of Figure 4 are repeating information in the legend to Figure 3 and can refer to it.

We revamped Figures 3 and 4. Figure 3 now shows the analysis of fibroblasts proteomics while Figure 4 focuses on neurons proteomics. We also modified the legend of Figure 4.

**Reviewer #2 (Recommendations For The Authors):**
(1) Abstract: 'Strikingly, TIMM50 deficiency had no impact on the steady state levels of most of its substrates, challenging the currently accepted import dogma of the essential general import role of TIM23 and suggesting that fully functioning TIM23 complex is not essential for maintaining the steady state level of the majority of mitochondrial proteins'. This sentence needs to be rephrased. The data do not challenge any dogma! The authors only show that lower levels of functional TIM23 are sufficient.

We have rewritten all the relevant sentences as suggested (details are also mentioned in response to reviewer 2 public review point 1)

(2) Introduction: 'Surprisingly, functional and physiological analysis points to the possibility that TIMM50 and a fully functional TIM23 complex are not essential for maintaining steady-state levels of most presequence-containing proteins'. This again needs to be rephrased.

Rewritten as suggested (details mentioned in response to reviewer 2 public review point 1)

(3) Discussion: 'In summary, our results challenge the main dogma that TIMM50 is essential for maintaining the mitochondrial matrix and inner membrane proteome, as steady state level of most mitochondrial matrix and inner membrane proteins did not change in either patient fibroblasts or mouse neurons following a significant decrease in TIMM50 levels.' This again needs to be rephrased.

Rewritten as suggested (details mentioned in response to reviewer 2 public review point 1)

(4) The analysis of the proteomics experiment should be improved. The authors show in Figures 3 and 4 several times the same volcano plots in which different groups of proteins are indicated. It would be good to add (a) a principal component analysis to show that the replicates from the mutant samples are consistently different from the controls, (b) a correlation plot that compares the log-fold-change of P1 to that of P2 to show which of the proteins are consistently changed in P1 and P2 and (c) a GO term analysis to show in an unbiased way whether mitochondrial proteins are particular affected upon TIMM50 depletion.

Figures 3 and 4 have been changed to avoid redundancy. Figure 3 now focuses on fibroblasts proteomics (with additional analysis), while Figure 4 focuses on neurons proteomics. PCA analysis was added in Fig S1, showing that the proteomics replicates of both patients (P1 and P2) are consistently different than the healthy control (HC) replicates. Correlation plots were added in Figure 3C and D, showing high correlation of the downregulated and upregulated mitochondrial proteins between P1 and P2. These plots further highlight that MIM proteins are more affected than matrix proteins and that the OXPHOS and MRP systems comprise the majority of significantly downregulated proteins in both patients. GO term analysis was performed for all the detected proteins that got significantly downregulated in both patients. The GO term analysis is displayed in Figure S3A, and shows that mitochondrial proteins, mainly of the OXPHOS and MRP machineries, are particularly affected.

(5) Figure 1. The figure shows the levels of TIM and TOM subunits in two mutant samples. The quantifications suggest that the levels of TIMM21, TOMM40, and mtHsp60 are not affected. However, from the figure, it seems that there are increased levels of TIMM21 and reduced levels of TOMM40 and mtHsp60. Unfortunately, in the figure most of the signals are overexposed. Since this is a central element of the study, it would be good to load dilutions of the samples to make sure that the signals are indeed in the linear range and do scale with the amounts of samples loaded.

The representative WB panels display the Actin loading control of the representative TIMM50 repeat (the top panel). However, each protein was tested separately, at least three times, and was normalized to its own Actin loading control.

(6) Figure 2B. All panels are shown in color except the panel for TIMM17B which is grayscale. This should be changed to make them look equal.

All the western blot panels were changed to grayscale.

(7) Discussion: 'Despite being involved in the import of the majority of the mitochondrial proteome, no study thus far characterized the effects of TIMM50 deficiency on the entire mitochondrial proteome.' This sentence is not correct as proteomic data were published previously, for example for Trypanosomes (PMID: 34517757) and human cells (PMID: 38828998).

We have corrected the statement to “Despite being involved in the import of the majority of the mitochondrial proteome, little is known about the effects of TIMM50 deficiency on the entire mitochondrial proteome.”

(8) A recent study on a very similar topic was published by Diana Stojanovki's group that needs to be cited: PMID: 38828998. The results of this comprehensive study also need to be discussed!!!

We have added the following in the discussion:

Line 362 – “These observations are similar to the recent analysis of patient-derived fibroblasts which demonstrated that TIMM50 mutations lead to severe deficiency in the level of TIMM50 protein (6,7). Notably, this decrease in TIMM50 was accompanied with a decrease in the level of other two core subunits, TIMM23 and TIMM17. However, unexpectedly, proteomics analysis in our study and that conducted by Crameri et al., 2024 indicate that steady state levels of most TIM23-dependent proteins are not affected despite a drastic decrease in the levels of the TIM23CORE complex (7). The most affected proteins constitute of intricate complexes, such as OXPHOS and MRP machineries. Thus, both these studies indicate a surprising possibility that even reduced levels of the TIM23CORE components are sufficient for maintaining the steady state levels of most presequence containing substrates.

(1) Homberg B, Rehling P, Cruz-Zaragoza LD. The multifaceted mitochondrial OXA insertase. Trends Cell Biol. 2023;33(9):765–72.

(2) Carroll J, Altman MC, Fearnley IM, Walker JE. Identification of membrane proteins by tandem mass spectrometry of protein ions. Proc Natl Acad Sci U S A. 2007;104(36):14330–5.

(3) Ting SY, Schilke BA, Hayashi M, Craig EA. Architecture of the TIM23 inner mitochondrial translocon and interactions with the matrix import motor. J Biol Chem [Internet]. 2014;289(41):28689–96. Available from: http://dx.doi.org/10.1074/jbc.M114.588152

(4) Trefts E, Shaw RJ. AMPK: restoring metabolic homeostasis over space and time. Mol Cell [Internet]. 2021;81(18):3677–90. Available from: https://doi.org/10.1016/j.molcel.2021.08.015

(5) Zhang S, Hulver MW, McMillan RP, Cline MA, Gilbert ER. The pivotal role of pyruvate dehydrogenase kinases in metabolic flexibility. Nutr Metab. 2014;11(1):1–9.

(6) Reyes A, Melchionda L, Burlina A, Robinson AJ, Ghezzi D, Zeviani M. Mutations in TIMM50 compromise cell survival in OxPhos‐dependent metabolic conditions . EMBO Mol Med. 2018;

(7) Crameri JJ, Palmer CS, Stait T, Jackson TD, Lynch M, Sinclair A, et al. Reduced Protein Import via TIM23 SORT Drives Disease Pathology in TIMM50-Associated Mitochondrial Disease. Mol Cell Biol [Internet]. 2024;0(0):1–19. Available from: https://doi.org/10.1080/10985549.2024.2353652